# Predicting maximum strain hardening factor in elongational flow of branched pom-pom polymers from polymer architecture

**Max G. Schußmann** [1], **Manfred Wilhelm** [1] & **Valerian Hirschberg** [1,2] ✉

We present a model-driven predictive scheme for the uniaxial extensional viscosity and strain hardening of branched polymer melts, specifically for the pom-pom architecture, using the small amplitude oscillatory shear mastercurve and the polymer architecture. A pom-pom shaped polymer is the simplest architecture with at least two branching points, needed to induce strain hardening. It consists of two stars, each with $q$ arms of the molecular weight $M_{w,a}$, connected by a backbone of $M_{w,b}$. Despite the pom-pom constitutive model, experimental data of systematic investigations lack due to synthetic complexity. With an optimized approach, we synthesized polystyrene pom-pom model systems with systematically varied $M_{w,a}$ and $M_{w,b}$. Experimentally, we identify four characteristic strain rate dependent regimes of the extensional viscosity, which can be predicted from the rheological mastercurve. Furthermore, we find that the industrially important maximum strain hardening factor depends only on the arm number by $[q^2/\ln(\sqrt{3}q)]$. This framework offers a model-based design of branched polymers with predictable melt flow behavior.

The quantitative connection between the molecular architecture (topology) and molecular dynamics is a long-standing, fundamental challenge in polymer science. The understanding and modelling of the flow of branched polymer melts is of great interest to both fundamental research and industrial applications like film blowing or foaming[1,2]. The pom-pom topology is the simplest branched topology with at least two branching points, which are needed to induce strain hardening. Strain hardening in elongational flow is the increase of the tensile stress growth coefficient above the linear viscoelastic envelope. Pom-poms consist of two star-shaped branch points connected by a linear backbone chain. Based on the tube model, McLeish and Larson developed constitutive equations to describe shear and extensional melt rheology of a pom-pom shaped polymer molecule[3,4]. The pom-pom model has been successfully applied to model the elongational flow behavior of low-density polyethylene (LDPE) and other long-chain branched (LCB) systems like combs or LCB-styrene-butadiene-rubbers[5–8]. Within the pom-pom model the stress is given by

$$\boldsymbol{\sigma}(t) = \frac{15}{4} G_0 \phi_b \left( \phi_b \lambda^2(t) + \frac{2qs_c(t)}{2qs_a + s_b} \right) \boldsymbol{S}(t) \text{ with } \phi_b = \frac{s_b}{2qs_a + s_b} \quad (1)$$

where $s_b$ and $s_a$ are the number of entanglements of the backbone and the arms respectively, $q$ is the arm number of each star of the pom-pom, $\phi_b$ is the volume fraction of the backbone, $\lambda(t) = [1, q]$ is the stretch parameter, $\boldsymbol{S}(t)$ is the orientation tensor, $s_c(t)$ is the branch point withdrawal parameter, and $G_0$ is the plateau modulus. The pom-pom model attributes two relaxation times to the backbone. The orientation relaxation time $\tau_b$, describing the alignment of the backbone tube due to flow, and the stretch relaxation time $\tau_s$, describing the stretch of the backbone chain in flow. In extension with $\dot{\varepsilon} > 1/\tau_s$, the backbone is stretched due to the friction of the branchpoints. At sufficiently high strains, the backbone reaches its

[1]Institute for Chemical Technology and Polymer Chemistry, Karlsruhe Institute of Technology (KIT), Engesserstraße 18, 76131 Karlsruhe, Germany. [2]Institute for Technical Chemistry, Technical University Clausthal, Arnold-Sommerfeld-Str. 4, 38678 Clausthal-Zellerfeld, Germany. ✉e-mail: valerian.hirschberg@kit.edu

maximum stretch $\lambda = q$ and thereafter the arms are retracted into the backbone tube, resulting in a steady state stress. The retraction of the arms is called branch point withdrawal and takes place at the time $t_q$ and strain $\varepsilon_q$ of maximum backbone stretch. Hierarchical relaxation of the arms before the relaxation of the backbone is essential in the pom-pom model. Therefore, the pom-pom model is limited to strain rates smaller than the inverse of the arm relaxation time $1/\tau_a$. The Rouse time of a polymer $\tau_R$, linear or branched[9,10], is the longest relaxation time of the molecule in the absence of entanglement effects; i.e., governed only by monomeric friction. This is the case if the applied elongational rate exceeds the inverse of the Rouse time. In the pom-pom model, strain hardening is the result of only branch point friction and at Hencky strain rates $\dot{\varepsilon} > 1/\tau_R$ all polymer chains independent of their topology show strain hardening due to monomer friction[11].

Macroscopically, the true strain or Hencky strain $\varepsilon$ for large deformations[1] is given by the initial length of the sample $L_0$ and the time-dependent length $L(t)$

$$\varepsilon(t) = \ln\left(\frac{L(t)}{L_0}\right) \tag{2}$$

From the time-dependent stress, the extensional viscosity $\eta_E$ at a constant strain rate $\dot{\varepsilon} > 1/\tau_s$, defined as the steady state of the time dependent tensile stress growth coefficient $\eta_E^+(t)$ in the pom-pom model and typically the maximum of $\eta_E^+$ with respect to time in the experiment, can be derived as

$$\eta_E = \frac{15}{4} G_0 \phi_b^2 q^2 \dot{\varepsilon}^{-1} = \frac{15}{4} G_0 \left(\frac{s_b}{2qs_a + s_b}\right)^2 q^2 \dot{\varepsilon}^{-1} \tag{3}$$

The Considère criterion (named after the French engineer *Armand Gabriel Considère*) states that a sample undergoes homogenous uniaxial elongation until the strain of the maximum force $\delta F/\delta\varepsilon < 0$[12–15]. Beyond the maximum stress, the material cannot be stretched homogenously and instead undergoes dynamical failure like necking or rupture. The Considère criterion is only valid for fast $\dot{\varepsilon}$ compared to the characteristic relaxation time $\tau$ of the material $\dot{\varepsilon}\tau \gg 1$. For pom-poms in this manuscript the inverse of the crossover frequency is used as the characteristic relaxation time, see discussion below. Corrections for Hencky strain rates where time and strain are relevant were evaluated by Fielding et al.[16,17]. McKinley and Hassager[18] used the Considère criterion and the pom-pom constitutive equations to predict the extensional viscosity of the pom-pom topology for rapid stretching. Rapid stretching is assumed for strain rates $\dot{\varepsilon} > 1/\tau_s$, where the viscous components can be neglected, and the strain energy is stored as elastic energy. In the following their main findings are shortly reviewed. $\eta_E^+$ normalized to the zero-shear viscosity $\eta_0$ can be expressed as a function of $q$, $t_q$, $\tau_b$, the Hencky strain to sample failure $\varepsilon_f$, and the strain of maximum backbone stretch $\varepsilon_q$ as

$$\log(\eta_E^+(t,\dot{\varepsilon})/\eta_0)|_{\varepsilon_f \to \varepsilon_q} = \log\left(\frac{3q^2}{\varepsilon_q}\right) + \log\left(\frac{t_q}{\tau_b}\right) \tag{4}$$

Here $t_q = \varepsilon_q/\dot{\varepsilon}$ and thus is dependent on strain rate. Equation 4 can be further simplified and is valid for the limiting case of $\varepsilon_f \to \varepsilon_q$. In the limit $\varepsilon_f \to \varepsilon_q$, the tensile stress coefficient can be simplified to the steady state extensional viscosity $\eta_E^+(t,\dot{\varepsilon}) \to \eta_E$ at a constant strain rate.

$$\frac{\eta_E}{\eta_0} = \frac{3q^2}{\varepsilon_q} \cdot \frac{t_q}{\tau_b} = \frac{3q^2}{\tau_b} \cdot \frac{1}{\dot{\varepsilon}} \tag{5}$$

This result may be compared with the linear viscoelastic envelope (LVE), for the experimental determination see experimental section,

approximated in the pom-pom model[3] by

$$\eta_{LVE}^+ = 3\eta_0 \left[1 - e^{\left(-\frac{t_q}{\tau_b}\right)}\right] \tag{6}$$

Which can for small times $\frac{t_q}{\tau_b} \ll 1$ be approximated with a first-order Taylor polynomial expansion, yielding a linear response

$$\eta_{LVE}^+ \simeq 3\eta_0 \left(\frac{t_q}{\tau_b}\right) \tag{7}$$

Rearranging Eq. (7) for $\left(\frac{t_q}{\tau_b}\right)$ and substituting it in Eq. (5) leads to

$$\eta_E \simeq \eta_{LVE}^+ \cdot \frac{q^2}{\varepsilon_q} \tag{8}$$

During the elongation, the strain of the sample increases until the $\varepsilon_f$ is reached. At the maximum stress, a force balance between the resistance against withdrawal of the branch point into the tube and further stretching of the backbone is reached. The maximum stress is obtained for $q \geq 2$ at $\lambda = q$, followed by a plateau for $\lambda > q$ (see Fig. 1 a). In the rapid stretching limit, the Hencky strain at failure can be derived from the stress tensor of the pom-pom model and is given by[18]

$$e^{2\varepsilon_f} + 2e^{-\varepsilon_f} = 3q^2 \tag{9}$$

through calculation of the force balance at maximum stretch by the pom-pom equations. For $q \geq 2$, $\varepsilon_q$ can be approximated using $e^{2\varepsilon_f} \gg e^{-\varepsilon_f}$ with

$$\varepsilon_q = \varepsilon_f \simeq \ln\left(\sqrt{3}q\right) \tag{10}$$

Combining Eq. (8) and Eq. (10) yields

$$\eta_E \simeq \eta_{LVE}^+ \cdot \left[\frac{q^2}{\ln\left(\sqrt{3}q\right)}\right] \tag{11}$$

predicting the increase of the extensional viscosity above the LVE by the Considère factor $f_c = [q^2/\ln(\sqrt{3}q)]$, see also Fig. 1 b). The Considère factor $f_c$ is directly related to the maximum achievable strain hardening factor $SHF_{max}$, see discussion. $SHF = \eta_E(\dot{\varepsilon})/\eta_{DE}(\dot{\varepsilon})$ and $\eta_{DE}$ is the steady state viscosity of the Doi-Edwards model. The relation between the Considère factor and the arm number is shown in Fig. 1 b), underlining the potential of high strain hardening through a high arm number. The experimental evaluation of the boundary conditions of the pom-pom constitutive equations are of high interest to verify the used simplifications. In addition, it allows the investigation of molecular dynamics of branched polymers in the melt state and the advancement of rheological modelling of more complex polymer topologies.

The relaxation dynamics of the arms and the backbone of the pom-pom are accumulated in the LVE. The linear viscoelastic envelope $\eta_{LVE}^+$ (LVE) is defined the complex viscosity, taking the Trouton ration, the Cox-Merz and Gleissle mirror rule into account.

$$\eta_{LVE}^+(t) = 3\eta(\dot{\gamma}^{-1}) \tag{12}$$

When a sample is subjected to sufficiently high, elongational stress or strain, it will undergo failure. Depending on the conditions, multiple failure mechanisms can take place. The pom-pom model states that the mechanism to reduce stress on the backbone is branch-point withdrawal at sufficiently high strains and strain rates, resulting in a steady-state viscosity. Herein, the tube of the backbone expands,

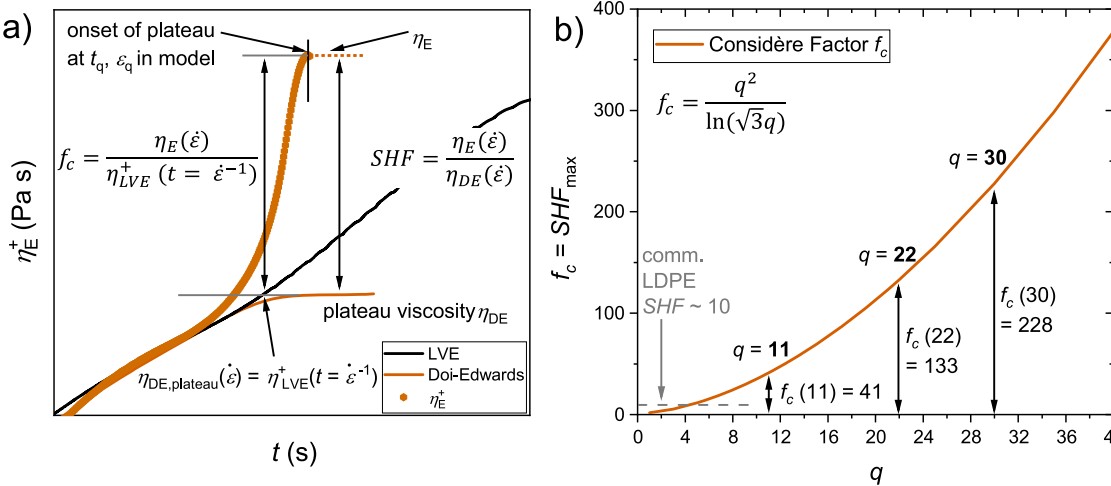

**Fig. 1 | Strain hardening in elongational flow and the Considère criterium. a** Schematic relations between $\eta_E$, $\eta_{DE}$, $\eta_{LVE}^+$, $f_c$ and $SHF$. The Considère factor is equal to the maximum strain hardening factor $SHF_{max} = f_c$, see discussion. **b** Relation of the Considère factor $f_c$ and the arm number $q$ of a pom-pom. Values for $q = 11, 22, 30$ are given as examples matching $q$ of the synthesized samples.

and the arms retract into the backbone tube[19,20]. At higher rates, the failure mechanism for linear chains was experimentally shown to be elastic rupture through chain scission for elastomers[21] and recently for polymer melts by chemoluminescence[22]. Similarly, a theoretical approach revealed the existence of a maximum stress of $\sigma_E^{Fracture}$ that a primary C-C bond can withstand before dissociation of the covalent bond. This was obtained by comparison of the C-C bond energy and the resulting strain energy on a Kuhn segment[23].

To date, rheological data of monodisperse, topologically defined LCB model-systems are rare[24–32], which limit progress in understanding the relation of molecular dynamics and non-linear viscoelastic rheology and therefore the comparison with constitutive equations[33]. The synthesis of pom-pom shaped model-systems with topological purity, low dispersity, and the systematic variation of the three topological parameters, backbone length $M_{w,b}$, number $q$ and length of the arms $M_{w,a}$, is a challenge to polymer chemistry. Many efforts have been made to develop synthetic routes using multifunctional linkers[34,35], converging synthesis using macromonomers and coupling methods[20,35–37]. All synthetic approaches yield the desired pom-pom architecture, but they lack precise control over molecular parameters of the pom-poms or are limited in one of them. For example, approaches with multifunctional linkers are typically limited to five arms per star because commercially available linkers contain up to six functional groups[34]. In addition, this approach is prone to crosslinking during the synthesis of the functionalized backbone due to the use of bi-anions and the multifunctional linkers. Moreover, convergent approaches lack precise control over the backbone molecular weight and are prone to contamination with symmetric and asymmetric stars. Recently a new facile synthesis method using a grafting-onto approach was developed, where all three molecular parameters, $M_{w,b}$, $M_{w,a}$, and $q$, are precisely controllable over a wide range[38–43]. Using this synthetic route, well defined pom-pom model systems can be synthesized in a 10 – 30 g range within five days[44].

In this work, we synthesized and investigate the impact of $M_{w,b}$ in oscillatory shear and extensional rheology of several pom-pom model systems and the comparison of the experimental data with the predictions of the pom-pom constitutive model. Their nonlinear rheological properties in uniaxial elongation is investigated and compared to the predictions of the pom-pom model of McLeish and Larson[3] and further simplifications of it by the Considère criteria by McKinley and Hassager[18]. The results show that for monodisperse pom-pom model systems the elongational viscosity as a function of the $Wi$ can be

entirely predicted via the knowledge of the topological parameters and the linear shear rheology. Experimentally, we find that the Considère factor is equal to the maximum strain hardening factor $f_c = SHF_{max}$. This finding allows the prediction of melt rheological properties and consequently paves the way towards a model-based design of branched polymer materials.

## Results

### Shear and elongational rheology

The SAOS mastercurves of the 24k-series are shown in Fig. 2a) for a reference temperature of $T_{ref} = 140\,°C$. For the pom-pom with $M_{w,b} = 100\,kg\,mol^{-1}$ only one rubber plateau, corresponding to the arm-arm and arm-backbone interactions, is observed. The backbone of the pom-pom with $M_{w,b} = 100\,kg\,mol^{-1}$ is not self-entangled, meaning the backbones of the pom-poms are not entangled with the backbones of other pom-poms at low angular frequencies, and therefore no backbone rubber plateau is shown. Distinct rubber plateaus for both the arms and the backbones can be seen for the samples with $M_{w,b} = 220\,kg\,mol^{-1}$ and $M_{w,b} = 400\,kg\,mol^{-1}$. Within these two later samples, the backbones are self-entangled. The rubber plateau of the backbone is shifted to lower moduli compared with linear chains due to dynamic dilution of the backbone with the arms. The separation of the rubber plateaus of the arms and the backbone is influenced by the difference in molecular weight und the volume fraction of arms and backbone. Higher differences in molecular weight between arms and backbone lead to a better separation between the two rubber plateaus. Volume fraction and molecular weight of the arms and the backbone will influence their respective minimum of the phase angle. Similar observations can be made for the 40k-series as shown in Supplementary Fig. 1a. In Fig. 2b, the phase angle $\delta$ is shown as a function of the angular frequency. Similarly to the mastercurves, the two self-entangled backbones of the high $M_{w,b}$ pom-poms can be identified. In these two samples, the phase angle of the low-frequency minimum goes to $\delta < 45°$, indicating the self-entangled backbones.

The tensile stress growth coefficients of the 24k-series as a function of time are shown in Fig. 2c at a reference temperature $T_{ref} = 140\,°C$. Selected strain rates are shown to illustrate the different uniaxial extensional behavior. At high Hencky strain rates ($\dot{\varepsilon} > 1\,s^{-1}$), the sample fails with elastic rupture[45]. This can be observed at a Hencky strain rate of $\dot{\varepsilon} = 3\,s^{-1}$ for 220k-2×12–25k and 400k-2×12–23k, visible by the sharp end of the tensile stress growth coefficient at Hencky strains smaller than $\varepsilon < 4$. At medium Hencky strain rates, the onset of

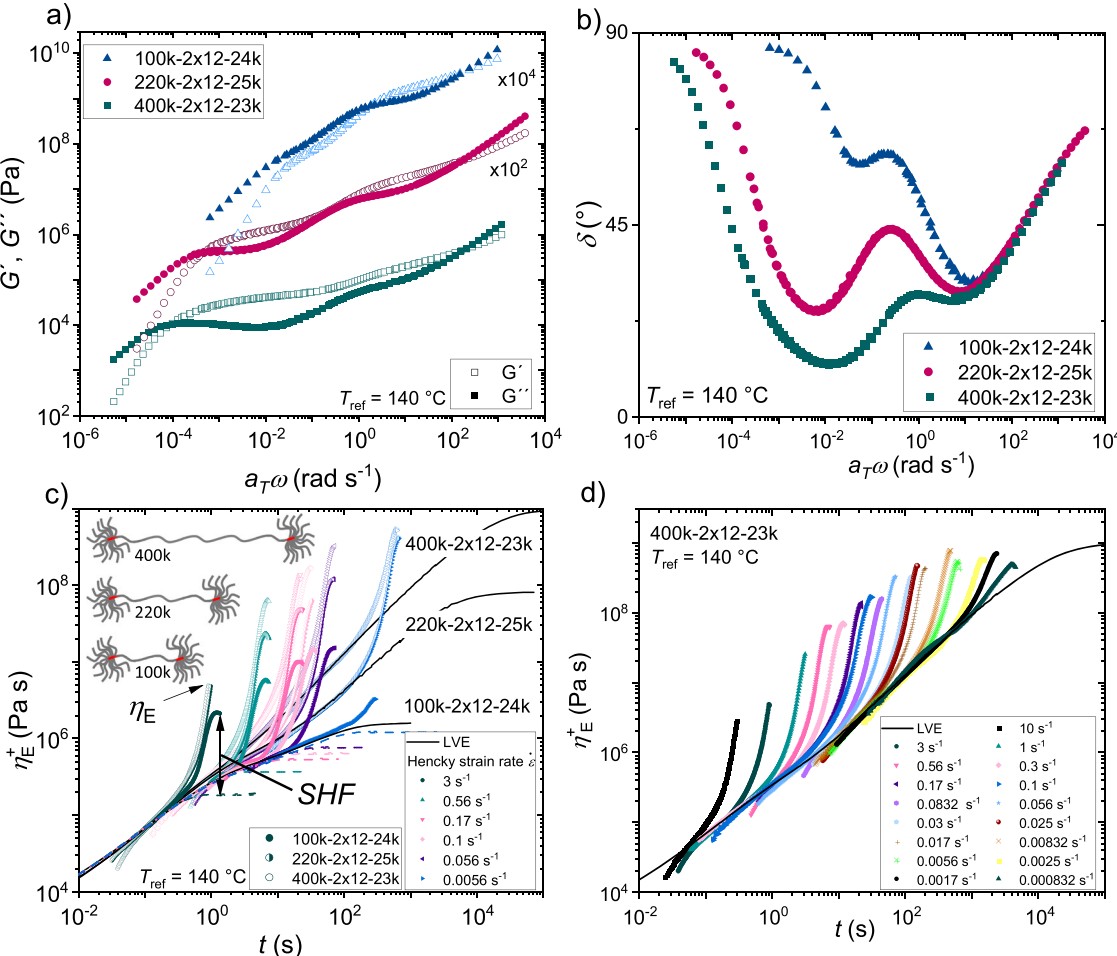

**Fig. 2 | Experimental shear and elongational rheological data of PS pom-pom model systems. a** Mastercurves of three pom-poms of the 24k-series. **b** Phase angle $\delta$ as a function of the angular frequency $a_T \omega$. **c** Tensile stress growth coefficient $\eta_E^+$ shown as a function of time of the 24k-series for selected strain rates. Dashed lines show the prediction of the Doi-Edwards model to illustrate the strain hardening factor *SHF*. **d** Tensile stress growth coefficient $\eta_E^+$ shown as a function of time for the pom-pom 400k-2x12-23k for all measured strain rates from 0.000832 to 10 s⁻¹. All figures are shown at a reference temperature of $T_{ref} = 140\,°C$.

the steady state viscosity is visible ($\dot{\varepsilon} = 0.56 - 0.056\,s^{-1}$). At low strain rates, the extensional viscosity approaches the linear viscoelastic envelope, and strain hardening substantially decreases. This can be seen at e.g., $\dot{\varepsilon} = 0.0056\,s^{-1}$ for the pom-pom 100k-2×12–24k. In addition, one can observe the high impact of the molecular weight of the backbone at small strain rates. Similar observation can be made for the 40k-series as shown in Supplementary Fig. 1.

It has been found in previous works on LDPE that at Hencky strains greater than $\varepsilon = 4$ and high strain rates, the tensile stress growth coefficient $\eta_E^+$ shows an overshoot before reaching steady state viscosity[46,47]. Due to the limitation of the EVF geometry to a maximum Hencky strain of $\varepsilon = 4$, the stress overshoot of polystyrene pom-poms remains an open question and needs to be investigated in the future with adequate equipment. In this work, not all samples at all strain rates reach a plateau in the stress below $\varepsilon = 4$. For these samples, the highest measured tensile stress growth coefficient value is considered the extensional viscosity.

**Constitutive modelling**

The experimental and via the pom-pom constitutive equation calculated tensile stress growth coefficients are shown as an example for the pom-poms 100k-2x11-9k and 400k-2×12–23k at a reference temperature of $T_{ref} = 140\,°C$ in Fig. 3 a) and b), respectively. The tensile stress growth coefficient of 100k-2x11-9k and all other samples found in Supplementary Fig. 2 are calculated based on the molecular

parameters determined from synthesis (see Supplementary Table 1) and PS specific rheological parameters ($\tau_e = 0.036\,s$, $G_0 = 2\,10^5\,Pa$). As shown in Fig. 3 a) as an example, the pom-pom model does not describe our experimental data well. Especially, the LVE of the pom-pom is not-well represented by the pom-pom model. As a result, the extensional data are shifted and are not matching well. In contrast, the strain hardening, so the increase of the tensile stress growth coefficient above the time dependent LVE, is relatively well predicted. Note that only volume fractions of $\phi_b = 0.09 - 0.42$ are investigated (also see Supplementary Table 1). The observed difference between experimental and calculated could be explained by the dominance of the arm relaxation behavior due to their high volume fraction resulting in backbones with no or only a small number of effective entanglements.

Pom-pom 400k-2×12-23k with the highest $\phi_b$ is shown in Fig. 3 b) with an increased $G_0 = 4\,10^5\,Pa$. With this increased modulus, experimental and simulated extensional data are in good agreement. Nevertheless, the plateau viscosity is underpredicted by factor of ~10, revealing different backbone relaxation times. The LVE can be represented by the pom-pom model if $G_0 = 1.8\,10^5\,Pa$, $\tau_e = 0.05\,s$ and $s_b = 38$ are chosen and therefore the backbone and entanglement relaxation times are artificially increased. So overall, the pom-pom model cannot represent the experimental LVE and elongational data correctly with the actual topological parameters, so the experimental data can only be predicted correctly if the molecular parameters are adjusted.

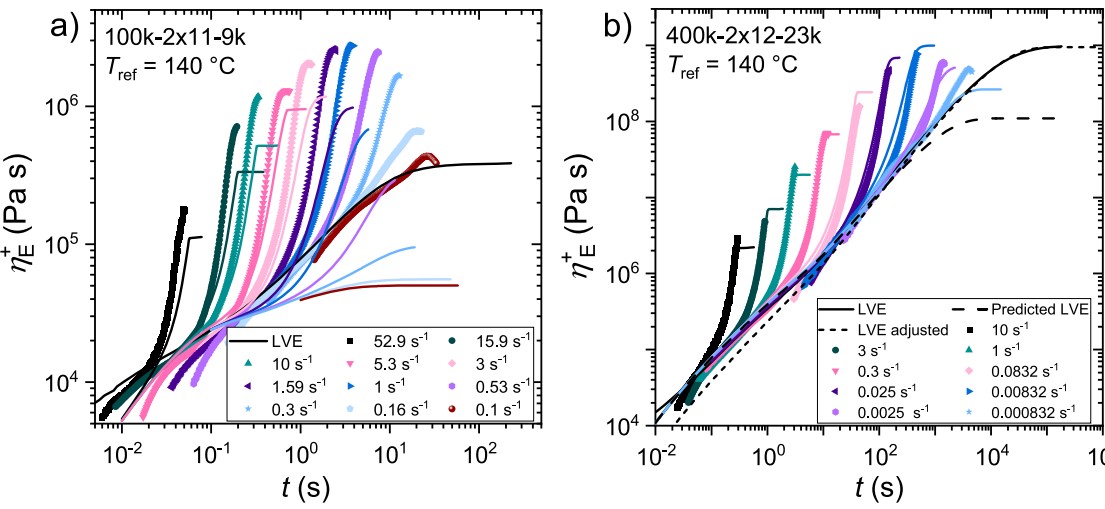

**Fig. 3 | Experimental tensile stress growth coefficient and the pom-pom model.** Tensile stress growth coefficient $\eta_E^+$ as a function of time for the pom-pom **a)** 100k-2×11-9k and **b)** 400k-2×12-23k at a reference temperature of $T_{ref} = 140$ °C. Dots show experimental data, solid lines show simulated tensile stress growth coefficient calculated by the pom-pom model with the respective molecular parameters and PS specific material parameters for **a)** and with adjusted $G_0 = 4\ 10^5$ Pa for **b)** for the solid coloured lines and the dashed black line. The dotted black line was obtained after fitting of the model to the experimental data obtaining $G_0 = 1.8\ 10^5$ Pa, $\tau_e = 0.05$ s and $s_b = 38$.

Contrary, the strain hardening prediction, i.e., the increase of the calculated tensile stress growth coefficient above the calculated LVE is roughly correct from the actual molecular parameters.

The Considère criterion states that a sample undergoes homogenous uniaxial elongation until the strain at which the maximum stress occurs. The application of the Considère criterion onto the pom-pom constitutive equations and the maximum theoretical fracture stress enables the prediction of the extensional viscosity of a given pom-pom shaped polymer sample. In Fig. 4, the extensional viscosity normalized to the zero-shear viscosity and the phase angle $\delta = \arctan(G''/G')$ of the SAOS mastercurve is shown as a function of the Weissenberg number $Wi = \tau_l \dot{\varepsilon}$ and Deborah number $De = \tau_l \omega$, respectively. $\tau_l$ is the longest relaxation time of a sample, determined by the crossover into the terminal regime. The longest relaxation time of the samples with not self-entangled backbones, namely 100k-2x12-24k and 100k-2x12-40k, is obtained by fitting a single mode Maxwell model to the terminal regime of G' and taking the crossover frequency of the Maxwell model as $\tau_l$. It appears that $\tau_l$ determined in this way is close to the stretch relaxation time $\tau_s$, see discussion below. For support, the mastercurve and the phase angle for similar samples are shown in Fig. 2 a) and b).

For pom-poms with self-entangled backbone, we can identify four regimes of the $Wi$-dependent elongational viscosity $\eta_E(Wi)$ and correlate it with the $De$-dependant shear relaxation behavior:

Regime I or the terminal/flow regime can be found at $Wi, De < 1$. In this regime, the backbone as well as the side chains are in their equilibrium stretch, and the extensional viscosity matches the LVE. Between ~ $0.3 < Wi < 1$, shear and extensional thinning can be observed, indicated by the decrease in the normalized viscosity below 1, caused by orientation of backbone chains due to the flow faster than the orientation relaxation time $\tau_b$. However, since the flow is slower than the stretch relaxation time $\tau_s$, no strain hardening can be observed here.

Increasing the strain rate applied to the sample, regime II or the transition regime can be found between $Wi, De = 1$ and the first minimum of the phase angle $Wi_{\delta,b,min}$. As $Wi, De > 1$, the backbone chains are stretched out of their equilibrium stretch due to $\dot{\varepsilon} > \tau_s^{-1}$. Therefore, an increase of the extensional viscosity above the LVE can be found, thus strain hardening is observed. The extensional viscosity increases with increasing Weissenberg number until the minimum of the phase

angle, where the highest extensional viscosities are measured. Similar to a ring-shaped sample, the crossover relaxation time of pom-poms seems roughly similar to the experimentally observed onset of strain hardening and therefore the stretch relaxation time[48]. The thinning observed in regime I indicates a longer orientation relaxation time. In contrast, Lentzakis et al.[9]. investigated elongational behavior of combs and derived a relation for the Rouse time of combs from linear chains by considering the additional monomeric friction contributed by the branches, giving

$$\tau_{R,br.} = \tau_e s_b \left(s_b + 2q s_a\right) \tag{13}$$

where $\tau_e$ is the relaxation time of an entanglement. For the investigated combs by Lentzakis et al., the Rouse time depends solely on the molecular parameters. To this point we are unaware of any exact equation for the prediction of the Rouse time of a pom-pom. We estimate that the Rouse time of a pom-pom should be slightly larger than the Rouse time of a comb due to its increased inertia at the chain ends. If Eq. (13) is applied to the pom-pom 280k-2x22-22k at $T_{ref} = 140$ °C and $\tau_e = 0.036$ s[49], a Rouse time for branched systems $\tau_{R,br.} = 44.6$ s can be calculated, which corresponds with $\tau_l = 1672$ s to $Wi(\tau_{R,br.}) = \tau_l/\tau_{R,br.} = 37.5$. Around $Wi = 37.5$, the pom-pom 280k-2x22-22k shows high strain hardening (regime III). The onset of strain hardening of the pom-pom 280k-2x22-22k and all other samples is around $Wi \sim 1$. Linear chains can relax by retraction of the chain ends, which determines the Rouse time and the onset of strain hardening. In well entangled linear systems, the Rouse time is related to the terminal relaxation time by $\tau_l/\tau_R \propto s$, where $s$ is the number of chain entanglements[50]. We assume, that the main stress in a pom-pom is generated on the backbone between the two branch points. Then, the retraction of an arm could yield no stress relaxation on the backbone due to the other arms retaining the position of the branchpoint and therefore the stress on the backbone remains. Stress release of a (partially) stretched backbone can in this picture only be achieved by movement of the whole branch point. This molecular hypothesis could explain the experimentally observed onset of strain hardening of the pom-poms at the inverse of the crossover frequency.

At higher Weissenberg numbers, regime III can be found between the phase angle minimum of the backbone and the onset of a plateau stress (see discussion below), $Wi_{\delta,b,min} < Wi < Wi_{\dot{\varepsilon}frac}$. In this regime, we

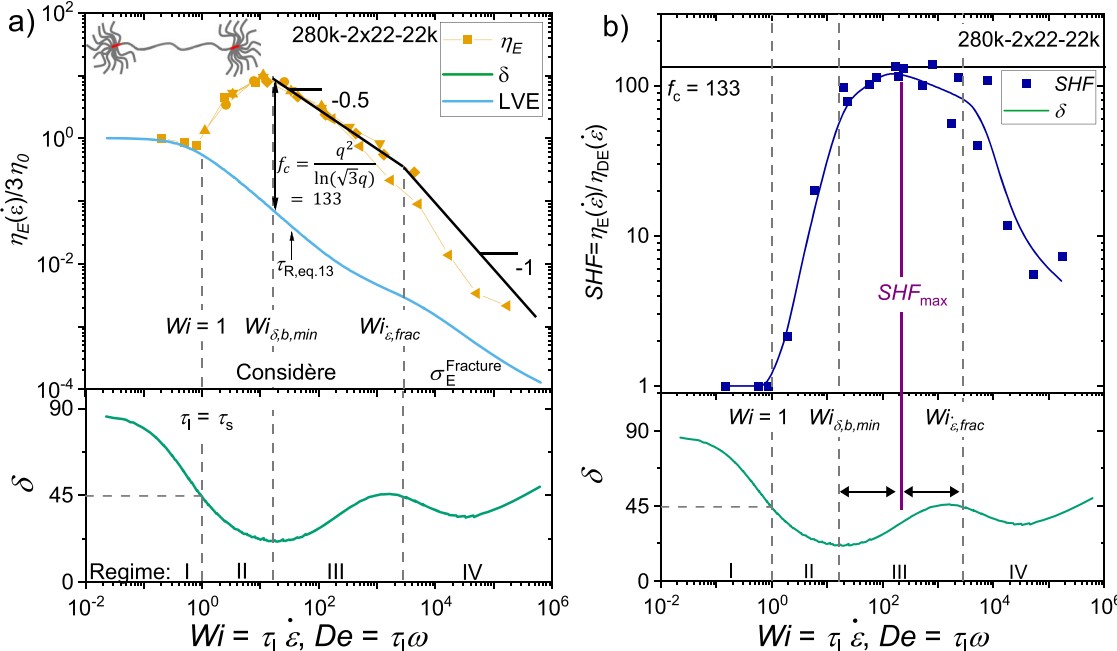

**Fig. 4 | Extensional viscosity, SHF and the phase angle from shear as a function of Deborah and Weissenberg number.** a) Extensional viscosity $\eta_E$ normalized by three times the zero-shear viscosity $\eta_0$ shown as a function of the Weissenberg number $Wi$ and the phase angle $\delta$ as a function of the Deborah number $De$ of the pom-pom 280k-2x22-22k. Different yellow symbols represent measurement temperatures: square 220 °C, circle 200 °C, upwards triangle 180 °C, downwards triangle 160 °C, rhombus 150 °C, and leftwards triangle 130 °C. Black lines are drawn as a guide for the eyes. b) Experimental strain hardening factor $SHF$ as a function of $Wi$ and $\delta$ as a function of $De$ of the pom-pom 280k-2x22-22k. The maximum reachable strain hardening factor can be predicted by assuming $SHF_{max} = f_c = 133$ for pom-pom topology. Blue line is as a guide to the eye.

experimentally find that the extensional viscosity (I) declines with a scaling exponent of about −0.5 and (II) matches the prediction of the Considère limit, a factor of $[q^2/\ln(\sqrt{3}q)]$ above the LVE as stated in Eq. (11). The assumption for Eq. (11) that $t_q/\tau_b \ll 1$ is justified at these $Wi$ numbers. For the pom-pom 280k-2x22-22k, this Considère factor $f_c$ is equal to $f_c = [q^2/\ln(\sqrt{3}q)] = 133$ for $q = 22$.

In regime III the arms of the pom-poms are still relaxed as in regime I and II, the backbone chains are both oriented and stretched during the elongation of the sample. Nielsen et al.[20]. studied elongational behavior of similar pom-pom shaped polystyrene melts with $q = 2.5$ and assume branch-point withdrawal into the backbone tube as the reason for the transition to steady stress which results in a steady-state viscosity at high Hencky strains. They investigated elongational behavior in the regimes I – III. Similarly, we find an onset of steady-state viscosity for our samples. Therefore, we propose branch-point withdrawal as a stress release mechanism in regime II to III in addition to stretch and orientation. Together with the Considère limit, these findings suggest that strain hardening (factor) $(SHF = \eta_E(\dot{\varepsilon})/\eta_{DE}(\dot{\varepsilon}))$ of pom-pom systems depends only on the number of arms attached to the branchpoints and not on the length of the arms or the backbone. This $SHF$ is given by the Considère limit as a function of the arm number. It will be investigated in the future at which length of the arms this will fail. The experimentally found scaling law of −0.5 is similar to previous findings on linear, branched, and ring systems[9,20,48,51,52]. In an ideal model, e.g., the pom-pom model, a slope of −1 is expected[3]. In the Doi-Edwards model, which can predict the rheology of linear chains, the slope is reduced to −0.8[53]. The slope in the experimental LVE and extensional viscosity is roughly -0.5. We think that this is a result of the superimposed relaxation behavior of the arms and the backbone leading to the decreased slope of -0.5 as well as a combination of chains reaching their stretch through finite extensibility and friction reduction.

Regime IV can be found at $Wi > Wi_{\dot{\varepsilon},frac}$. The elongation of the sample takes place until elastic rupture of the sample. We find

experimentally a limiting fracture stress of $\sigma_E^{Fracture} = 2 \; 10^7 \, \mathrm{Pa}$ (see Supplementary Fig. 11) and therefore resulting in a scaling exponent of −1 for the extensional viscosity. $Wi_{\dot{\varepsilon},frac}$ is obtained from the onset of the stress plateau indicated by the change in power law from 0.5 or 1 to plateau stress in stress vs. strain rate, see Supplementary Fig. 12a−d). A similar limiting fracture stress was predicted theoretically by Wagner et al.[23]. Due to the independence of the sample failure from the strain rate, we assume that the fracture of the covalent C-C bonds is the main stress release mechanism for pom-poms in regime IV, similar to linear chains[22]. $\dot{\varepsilon}_{frac}$ is found to be a function of the span molecular weight $M_{w,span} = M_{w,b} + 2M_{w,a}$ at constant $q$ and scales with a power law of −2.4.

With these findings the extensional viscosity over the complete $Wi$ range of a given pom-pom sample with known molecular parameters can be predicted solely from the LVE. Starting at high $Wi$, the extensional viscosity is limited by the fracture stress (scaling −1) until $Wi = Wi_{\dot{\varepsilon},frac}$. Between $Wi_{\delta,b,min} < Wi < Wi_{\dot{\varepsilon},frac}$, the extensional viscosity is limited by the Considère limit and scales with a scaling exponent of −0.5. For $Wi < Wi_{\delta,b,min}$, the extensional viscosity decreases to the LVE and matches the LVE for $Wi < 1$. With this procedure, the extensional viscosity can be predicted over the whole Weissenberg number range, with only the LVE and the topological parameters of the given pom-pom molecule. Furthermore, it might be possible to invert this result. With the combination of the Considère factor $(f_c = [q^2/\ln(\sqrt{3}q)])$ and the LVE data, the average arm number of an analogous pom-pom molecule for low disperse branched systems can be determined. This could be achieved simply by adjusting the arm number of a model pom-pom to match the experimental extensional viscosity via the Considère limit and the approximations that resulted in Eq. (11).

The four $Wi$ dependent regimes identified with the pom-pom 280k-2x22-22k can also be found in the other pom-pom samples with arm lengths of $M_{w,a} \geq M_e/2$. They are shown in Supplementary Fig. 3 to Supplementary Fig. 9.

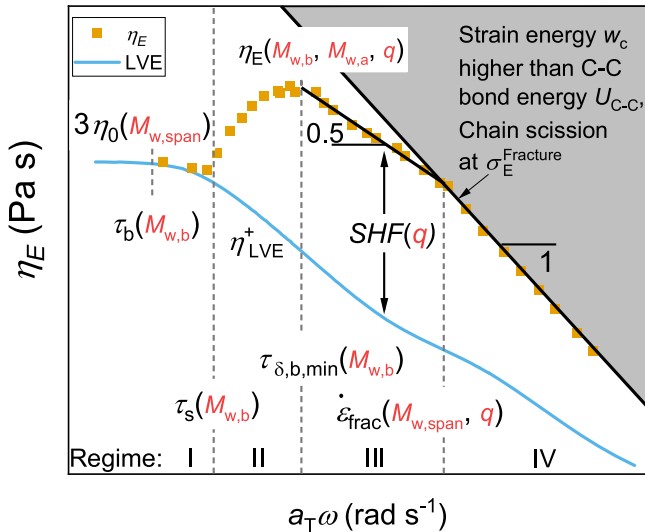

**Fig. 5 | Mapping elongational viscosity as a function of frequency.** Generalised relationships for the strain hardening depending on the molecular topology of pom-poms with entangled arms and backbone are shown to illustrate the overall trends, allowing to design pom-pom molecules according to desired properties. Idealized LVE and extensional viscosities in the light blue line and the orange squares, respectively.

For the sample 280k-2x30-7k, no scaling of −0.5 and no regime III was experimentally found in the extensional viscosity. The extensional viscosity matches the Considère prediction only at one point around $Wi_{\delta,b,\min}$, as shown in Supplementary Fig. 10. In agreement, only one minimum can be found in the phase angle. The vanishing of the third regime occurs around $M_{w,a} \sim M_e/2$ as shown by the samples 100k-2×11-9k and 280k-2x30-7k. Further investigations of very short arms should be conducted in the future. The samples 100k-2×12-24k and 100k-2x12-40k show no self-entangled backbone as shown in Fig. 2 and Supplementary Fig. 1. The prediction of the Considère limit overestimates the extensional viscosity as shown in Supplementary Fig. 4 and Supplementary Fig. 5, respectively. The pom-pom model assumes the backbone to be entangled with other backbones for the constitutive equations to be valid. This is not the case for these two samples. The dynamics of not self-entangled backbones needs to be investigated in the future.

The strain hardening factor ($SHF$) is widely used in industry to easily quantify and compare the strain hardening of polymer melts of different architectures and chemistries[53]. The strain rate-dependent $SHF(\dot{\varepsilon})$ of a material can be summarized into the rate-independent, maximum $SHF$ of a material $SHF_{\max}$. In Fig. 4 b) we show the $SHF$ as a function of $Wi$ for the pom-pom 280k-2x22-22k. Similar to Fig. 4a), the four regimes can be identified in the $SHF$. We experimentally find that the maximum strain hardening factor and the Considère factor are identical $SHF_{\max} = f_c$ within the experimental error for pom-poms with $M_{w,a} \geq M_e/2$. $SHF_{\max}$ is found in regime III between $Wi_{\delta,b,\min}$ and $Wi_{\dot{\varepsilon},\text{frac}}$. Note that the Considère factor $f_c$ is normalized against the LVE $f_c = \frac{\eta_E(\dot{\varepsilon})}{\eta_{LVE}^+(1/t)} \simeq \left[\frac{q^2}{\ln(\sqrt{3}q)}\right]$. When comparing the plateau viscosity and the steady state viscosity of the DE model, as shown in Supplementary Fig. 16, one can observe $\eta_{LVE}^+(1/t) = \eta_{DE}(\dot{\varepsilon})$ within the uncertainty of the experiment. $SHF_{\max} = f_c$ can therefore directly be derived from their definitions above.

For the pom-pom 280k-2x30-7k, $SHF_{\max} \sim 0.5 f_c$ is found near $Wi_{\delta,b,\min}$. The Considère factor of three pom-poms with arm numbers of $q = 11, 22, 30$ are shown exemplary in Fig. 1b). For comparison, commercial LDPE typically has moderate strain hardening with $SHF_{\max}$

around $10^{54-58}$. A pom-pom analogue polymer chain would only need around five arms per branch point, 10 in total, to reach similar $SHF_{\max}$, underlining the immense potential of the pom-pom topology to tune strain hardening e.g., to improve mechanically recycled polymer melts. The found relation between $SHF_{\max}$ and $f_c$ enables the development of materials with drastically increased strain hardening based on the simple insight that more arms cause more strain hardening if the backbone is self-entangled.

Generalised molecular topology – strain hardening relationships of pom-poms with entangled arms and backbone are shown in Fig. 5. The four $Wi$, respectively frequency dependent regimes are defined by three distinct relaxation times. The onset of the strain hardening between regime I and II is given by the stretch relaxation time $\tau_s$ and depends on the backbone molecular weight $M_{w,b}$. The range of regime III is defined by the separation between the phase angle backbone minimum $\delta_{b,\text{min}}$ and onset of the plateau stress. The phase angle minimum is dependent on $M_{w,b}$. The strain hardening in regime III is only a function of the number of arms, as discussed earlier. More arms result in higher strain hardening. In regime IV, the strain hardening is limited by chain scission at the fracture stress. This topology-strain hardening map allows the design of pom-pom molecules with a specific $SHF$ at a specific $\dot{\varepsilon}$ and $\eta_0$.

Besides the pom-pom topology, there are other LCB topologies like combs or Cayley-trees. Since pom-pom-like constitutive equations have been proposed for combs[6], the question presents itself whether the Considère limit can be applied to other topologies as well. Data reported by Abbasi et al. on combs[49,59] indicates that the extensional viscosity can be predicted by the Considère limit for combs with entangled backbones. Half of the total amount of arms of the comb is used for $q$. In the limiting case of very loosely grafted combs (average three arms or $q = 1.5$) the Considère limit fails, as shown in Supplementary Fig. 14 and Supplementary Fig. 15. This correlation between defined but less branched comb architectures and the pom-pom constitutive model predictions should be evaluated in further work. Furthermore, rheological investigations and synthesis should be focused on topologies that can surpass the Considère criterion with $SHF_{\max} > [q^2/\ln(\sqrt{3}q)]$. The topologies might need to be more complex than two multi-arm branch points as in a pom-pom or many single-arm, threefold branch points as in combs. In addition, we analysed the extensional data from Nielsen et al.[20]. of a pom-pom with $M_{w,b} = 14$ k mol$^{-1}$, $M_{w,a} = 27$ kg mol$^{-1}$, and an average of $q \sim 2.5$. The Considère factor ($f_c \sim 4.3$) underpredicts the extensional viscosity similarly to loosely grafted combs. We hypothesize that the underprediction might be due to the low arm number and therefore maybe a different relaxation mechanism is dominant.

As stated in Eq. (3) from the pom-pom model, the extensional viscosity $\eta_E$ is predicted to scale quadratically with the backbone volume fraction. For the 24k-series, we show the extensional viscosity of different strain rates as a function of the backbone entanglement number in Fig. 6a) and find a scaling exponent of 2. As predicted by dilution theory and experimentally observed by Abbasi et al.[49]. for PS comb polymers, the dilution modulus $G_{N,s}^0$ is related to the plateau modulus $G_0$ of linear polystyrene by $G_{N,s}^0 = G_0\phi_b^2$. Similar observations could be found for pom-poms with $\phi_b^2 > 0.1$ by Hirschberg et al.[60]. In Fig. 6b), the extensional viscosity normalized to the dilution modulus is shown as a function of the backbone volume fraction. Consequently, Eq. 2 can be rewritten as

$$\eta_E = \frac{15}{4}\frac{G_0\phi_b^2 q^2}{\dot{\varepsilon}} = \frac{15}{4}\frac{G_{N,s}^0 q^2}{\dot{\varepsilon}} \tag{14}$$

We find that the extensional viscosity normalized to the dilution modulus $\eta_E/G_{N,s}^0$ is independent of the backbone molecular weight for

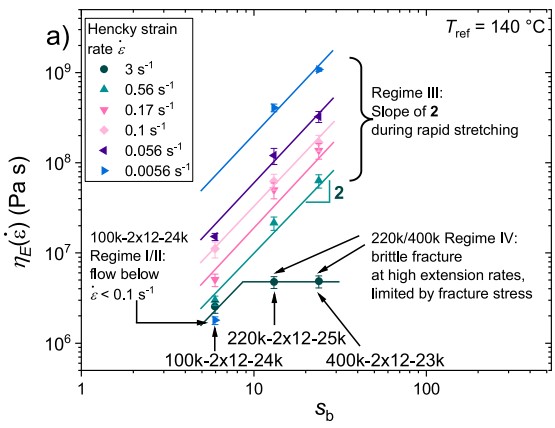

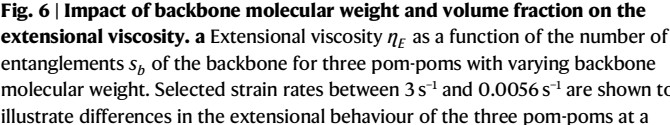

**Fig. 6 | Impact of backbone molecular weight and volume fraction on the extensional viscosity. a** Extensional viscosity $\eta_E$ as a function of the number of entanglements $s_b$ of the backbone for three pom-poms with varying backbone molecular weight. Selected strain rates between 3 s⁻¹ and 0.0056 s⁻¹ are shown to illustrate differences in the extensional behaviour of the three pom-poms at a

reference temperature of $T_{ref}$ = 140 °C. At $\dot{\varepsilon}$ = 3 s⁻¹ the extensional viscosity is limited by the maximum fracture stress. **b** Extensional viscosity normalized to the dilution modulus $\eta_E/G_{N,s}^0$ shown as a function of the volume fraction of the backbone $\phi_b$. Error bars represent typical errors in the experimental measurement of the extensional viscosity.

the pompoms with $M_{w,b} > 100$ kg mol⁻¹ and self-entangled backbone. Similar findings in extension are obtained for the 40k-series, as shown in Supplementary Fig. 17.

## Discussion

We have synthesized and investigated the rheological behavior of multiple polystyrene pom-pom model systems with systematically varying backbone molecular weight $M_{w,b} = 100–400$ kg mol⁻¹, arm molecular weight ($M_{w,a} = 24$ or $40$ kg mol⁻¹), and similar number of arms ($q_{24k} = 12$, $q_{40k} = 10–13$). Their melt rheological response is analysed in steady-rate uniaxial extension with respect to the equations of the constitutive pom-pom model and the Considère criterium as a function of the Weissenberg number $Wi = \tau_i \dot{\varepsilon}$, and correlated with the relaxation behaviour in small amplitude oscillatory shear. We found that the pom-pom-like behaviour is dominant within the melt for all pom-poms due to the overlapping extensional viscosities $\eta_E$. We identified four $Wi$ dependent, characteristic regimes in the extensional behavior of the pom-pom melts. Whereas no strain hardening is found in regime I for $Wi < 1$, regime II ($1 < Wi < Wi_{\delta,b,min}$) shows the onset of strain hardening. $\eta_E$ is increasing up to its maximum at the transition to regime III, the Weissenberg number of the first low frequency phase angle minimum, corresponding to the backbone, $Wi_{\delta,b,min}$. In regime III ($Wi_{\delta,b,min} < Wi < Wi_{\dot{\varepsilon},frac}$), it was experimentally found that $\eta_E$ scales with a scaling law of −0.5 as a function of $Wi$. The Considère limit yields accurate predictions of $\eta_E$ of the pom-pom samples with an increase of $\eta_E$ by the Considère factor $f_c = \left[q^2/\ln\left(\sqrt{3}q\right)\right]$ above the linear viscoelastic envelope at $Wi_{\delta,b,min}$. The Considère factor is found to be directly related to the maximum strain hardening factor $f_c(q) = SHF_{max}$ ($SHF = \eta_E(\dot{\varepsilon})/\eta_{DE}(\dot{\varepsilon})$) consequently the strain hardening is only depending on the number of arms. In regime IV ($Wi > Wi_{\dot{\varepsilon},frac}$), the extensional viscosity was found to scale with −1 as a function of $Wi$ due to being limited by a maximum fracture stress of $\sigma_E^{Fracture} = 2\,10^7$ Pa, as higher values lead to a breaking of covalent C-C bonds within the main chain. With these findings, one can predict the extensional viscosities and therewith the $SHF_{max}$ via the pom-pom constitutive equations over the whole Weissenberg range knowing only the molecular topological parameters and the linear shear relaxation behavior. Furthermore, we showed that the Considère limit can also be applied to combs with medium to high grafting densities if half of the total number of branches is considered as $q$. For the pom-poms, we find that the extensional viscosity normalized to the solution modulus $\eta_E/G_{N,s}^0$ is

independent for the backbone molecular weight for pom-poms with self-entangled backbones.

## Methods

All samples were investigated in small amplitude oscillatory shear (SAOS) and steady Hencky strain rate uniaxial elongational flow. Both types of rheological experiments were conducted on an ARES-G2 rheometer (TA Instruments, Newcastle, USA) using a 13 mm plate-plate geometry and an extensional viscosity fixture (EVF) under nitrogen atmosphere. Samples were hot-pressed at 180 °C for 15 min under vacuum to avoid degradation. Shear measurements were conducted between 110 and 260 °C and an angular frequency range of $\omega = 0.05–100$ rad s⁻¹. Elongational measurements were conducted between 130 °C and 220 °C and in a Hencky strain rate range of $\dot{\varepsilon} = 0.03–10$ s⁻¹. The measurement of the tensile stress growth coefficient on the EVF geometry is limited to a Hencky strain of $\varepsilon \leq 4$. Generally, for $q \geq 22$, at high rates the sample fails at $\varepsilon < 4$, at medium and low strain rates the sample does not fail until $\varepsilon > 4$. However, the stress plateau and therefore the viscosity plateau is typically reached for $\varepsilon < 4$. For $q \leq 13$, samples fail $\varepsilon < 4$ at all strain rates. The linear viscoelastic envelope (LVE) is obtained through the measurement of the complex viscosity in SAOS, the Cox-Merz relation and the Trouton ratio (Eq. 12)[61-63].

A list of symbols is included in the supplementary information.

## Data availability

The data that support the findings of this study are available from the corresponding author upon request.

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

## Acknowledgements

We thank Prof. Manfred Wagner for fruitful discussions and Prof. H. Henning Winter for access to IRIS Rheo-Hub software. Discussions with Dr. Hyeong Yong Song are gratefully acknowledged.

## Author contributions

M.G.S. and V.H. synthesized and characterized the samples. M.G.S. and V.H. performed rheological measurements. All authors (M.G.S., M.W., V.H.) contributed to the writing of the paper. Project conceived by V.H.

## Funding

## Competing interests

The authors declare no competing interests.
