## [Peer Review File · Nature Communications]

Predicting Maximum Strain Hardening Factor in Elongational Flow of Branched Pom-Pom Polymers from Polymer ArchitectureReviewers' Comments:

Reviewer #1:

Remarks to the Author:

This work analyzes the extensional rheology for a series of Pom-Pom polystyrenes with well-defined molecular architecture, and links the rheological behavior especially the maximum strain hardening factor to the molecular topology. Extensional rheology data of model branched polymer is rare. This work provides valuable experimental data that improves our understanding of polymer dynamics in strong flow. In particular, the generalized relationship between the strain hardening and the molecular topology of Pom-Poms in Fig.6 of the paper is very interesting and provides an important guidance for molecular design of branched polymers. However, there are also a few points requiring further clarification from the authors before the work can be published:

1. The samples are all measured at the same reference temperature, but do they have the same glass transition temperature T_g ? Do the short arms influence T_g ?

2. In Fig.3a, the sample 100k-2x12-24k shows only one rubber plateau, and the authors relates this plateau to entangled arms and concludes that the backbone is not entangled. But I think this plateau should be due to entanglements including arm-arm, arm-backbone, and backbone-backbone interactions. How do the authors know this plateau is from arm-arm entanglements only? Furthermore, in Fig.3b, what does the low-frequency minimum of 100k-2x12-24k mean (although $G'' > G'$)?

3. In Fig.4a, the predictions of the pom-pom model underestimate the steady state viscosity severely especially at fast rates. However, I am wondering if the authors checked the real Hencky strain of the measured samples. For example, the simulations from J. Non-Newtonian Fluid Mech. 165, 14–23 (2010) show that for highly elastic materials stretching on SER, the actual Hencky strain can deviate from the ideal Hencky strain significantly (also depends on initial sample size). When the ideal Hencky strain is 2, the actual Hencky strain of the sample can be as low as less than 1. That means the actual cross-section area is much larger, and the corresponding stress should be lower. Stretching samples on EVF may have the same problem, especially for the samples with high strain hardening. Therefore, I think the authors should check the cross-section area at the end of stretching for at least one sample (which shows high strain hardening and deviates significantly from pom-pom model prediction).

4. In Eq.4, it would be helpful if the authors can further clarify the difference between “the Hencky strain-to-failure” and “the maximum Hencky strain” – why the Hencky strain-to-failure is not the “maximum” Hencky strain?

5. From Eq.11, f_c is defined as the ratio of the extensional viscosity and the corresponding startup viscosity on the LVE; but on page 5 line 119, f_c is equal to the ratio of the extensional viscosity and the steady state viscosity of the Doi-Edwards model. This is a little bit confusing - are the two definitions equivalent to each other?

6. On page 13 line 270, the authors mentioned that “ τ_{-1} is the longest relaxation time of a sample, determined by the crossover into the terminal regime”. Is τ_{-1} of the 100k backbone sample also determined the same way? From Fig.3a, the crossover into the terminal regime of 100k-2x12-24k is apparently not corresponding to the longest relaxation time.

7. On page 16, line 334-337, the authors mentioned that “the extensional viscosity is only depending on the backbone length” and “strain hardening depends not on the length of the backbone.” It is not very straightforward to understand and would be helpful if the authors can explain it further.

8. In Fig.7b, would it be better to use the extensional stress (instead of the extensional viscosity) to normalize with the plateau modulus?

Reviewer #2:

Remarks to the Author:

Here the authors study linear rheology and extensional flow behaviour of pom-pom type polymers, synthesised to have a large number of arms at the end of the polymer. They analyse the behaviour in terms of the pom-pom model of McLeish and Larson, making use of a Considere criterion for the elongation at break.

Unfortunately a lot of the analysis is performed in a way that betrays a misunderstanding of the pom-pom model and the expected molecular relaxation process, and the limitations of the model. So, I think quite a lot of the analysis is flawed or incomplete. I'll try to work through the paper detailing places where more consideration is really needed.

1) In the introduction, leading to Eqs 1-11, more detail is needed, especially on what assumptions are made along the way and what limits are being taken. For example, assumptions are made as to strain rate and strain in order to find eq 3 as a limiting case from eq 1. For eq 4 it is not really explained what the difference is between strain to failure and maximum strain; again in eq 9 there are assumptions made about the stretching rates. These all need detailing otherwise the reader needs to look back at the original papers and figure it out for themselves.

2) Also eq 12 is not clear.. the right hand side is given in terms of angular frequency (ω) which seems odd in context. Surely we want a function of time?

3) In the methods section on rheology, it should be discussed what the maximal extensional strain of the instrument is, and whether it is ever reached before breaking (or does the sample always break first?)

4) Figure 3c (and other similar figures in the main body of the paper) only really shows the highest extension rates (I think typically regions III and IV from the later analysis), with the full range of extension rates only in the supplemental info. I think it's important to see at least one instance of the full range of rates.

5) Figure 4: it's important to detail what parameters were used for the pom-pom predictions, especially what values of orientational and stretch relaxation times. The orientation time should be close to the longest relaxation time. Looking at the extended predictions in SI, it seems that often the LVE at long times is underpredicted by the parameters used, ... but this is where the predictions should be matched because the long times correspond to the backbone relaxation. Matching at intermediate times around $t=10$ corresponds to angular frequencies of ~ 0.1 in the LVE, which is nowhere near the backbone orientation time. Hence I think the pom-pom fitting may not be done correctly.

6) page 13, There is a confusing sentence: "For the sample 280k-2x30-7k, the experimental and calculated tensile stress growth coefficients 256 are shown in Figure 4b). Overall, the tensile stress growth coefficient is predicted more 258 accurately than for samples with unentangled arms (280k-2x30-7k...." The two occurrences of 280k-2x30-7k don't make sense together!

7) Page 13: it is worth noting that the crossover time in LVE is not always the best measure of longest relaxation time, especially when a lot of dynamic dilution has taken place. Likely the longest relaxation time is a bit longer than estimated here.

8) Page 14: "As $Z[\omega]H > 1$, the backbone chains are not fully relaxed, and the backbone polymer chain can be partially both oriented and stretched during elongation...." and likewise page 15: "terminal

relaxation time of pom-poms is equal to the experimentally observed onset of strain hardening." and "Stress release of a (partially) stretched backbone can in this picture only be achieved by the reptation of the whole pom-pom molecule."

Here the authors have not understood the difference between the orientation relaxation time and stretch relaxation time in the pom-pom model. Typically the orientational relaxation time is the longest relaxation time in LVE, and for flow rates greater than inverse of orientation time some extension thinning is expected because tubes orient but chains do not stretch. This is seen to the left of the $Wi=1$ line in fig 5 (an indication that the longest relaxation time is a little underestimated). Once the flow rate is greater than the inverse of the stretch time then chains stretch and extension hardening is seen. For backbones with only a few (diluted) entanglements, as in 280k-2x22-22k, the stretch and orientation times are quite close, but they are not the same thing.

9) Eq 9: it is a good idea to estimate the Rouse time as with equation 13, which is for combs. Likely the Rouse time of a pom-pom will be a bit larger than this, because for a comb the monomer friction from arms is distributed along the backbone, whilst for the pom-pom it's all at the chain ends. There is some confusion about τ_e ... it's given as 0.036s on page 15 and 0.225s on page 7. I think the 0.036s is probably closer, but I'd estimate a bit bigger than that. In any case, for flow rates above the inverse of the Rouse time, the pom-pom model no longer applies because at these rates the monomer friction alone is sufficient to stretch the chains and branch-point withdrawal (which happens at the rate of the Rouse time) can no longer occur. The pom-pom model assumes flow rates smaller than inverse of Rouse time. Here, then, we expect chains to continue stretching until maximal extension.

Unfortunately this means that the pom-pom model should not be applied at all at these high rates, which corresponds to regimes III and IV. So, in these regions, the Considere criterion derived from the pom-pom model should not apply.

I suspect that what is happening in practice here is that there is a combination of chains reaching their maximal stretch through finite extensibility and of the reduction of chain friction (as seen in oriented linear chains). This gives rise to a -0.5 slope for strongly stretched linear polystyrene chains, and I think it's the same thing happening here.

10) Page 15: "Equation (13) seems to systematically overestimate the Weissenberg number of the strain hardening onset, suggesting a difference in the relaxation mechanism for combs and pom-poms."

This sentence betrays a fundamental misunderstanding of two different stretch relaxation times in branched polymers (pom-poms or combs). The "stretch relaxation time" in the pom-pom model is derived by considering that the branch-points take hops at the timescale of entangled arm relaxation.. this hopping gives rise to an effective friction from the hopping motion, and the stretch relaxation time is found by balancing the hopping friction with the backbone spring constant. If the flow rates are faster than this stretch time, the pom-pom model predicts the backbone stretches up to the maximum stretch given by the priority, at which point branchpoint withdrawal occurs. There is a directly analogous stretch relaxation time for comb polymers (again from hopping of the branchpoints, and again with the possibility of branchpoint withdrawal) as described in ref 44. The Rouse time, on the other hand, is just the stretch relaxation time for the backbone pulling against MONOMER friction (not branchpoint hopping friction). As detailed above, if the flow is faster than the Rouse time then chains stretch and do not even have time for branchpoint withdrawal. Clearly the Rouse time is shorter than the pom-pom stretch relaxation time.

11) For the reasons given in point 9 and 10 above, the main conclusion for region III is wrong, e.g as described in page 15:

"At higher Weissenberg numbers, regime III can be found between the phase angle minimum of 321 the backbone and the relaxation time of the arms, $Z[\nu, \square]$, $wx < Z[< Z[J, \square]$. In this regime, we

322 experimentally find that the extensional viscosity (I) declines with a scaling exponent of about
323 -0.5 and (II) matches the prediction of the Considère limit, a factor of $[\square]$

}ln $\sqrt{3}$] above the
324 LVE as stated in equation (11)."

Simply put, you can't use the Considere criterion derived from the pom-pom model at flow rates outside the regime of validity of the pom-pom model.

12) Finally, to regime IV, described from page 16 onwards. Here we are at rates much faster than the Rouse time... indeed the rates are approaching the entanglement time of PS. This happens coincidentally to be a similar order of magnitude to the arm relaxation time, here because the arms are quite short, around an entanglement length long. So, really the physics here is that ALL the molecule is stretching on scales around an entanglement length, arms, backbone, everything...there is not even time to redistribute monomers along the backbone tube. At these rates it is no surprise that chains might get stretched enough to break, although I suspect the mode of breaking is more subtle than described in ref 17.

Overall: I think there are serious misunderstandings of the molecular physics and of the pom-pom model displayed in the analysis. I think this means the conclusions are quite flawed.

Having said all that: the data is really good and valuable.. There are measurements of these samples across a broad range of extension rates and that would be an excellent resource to the community.

Reviewer #3:

Remarks to the Author:

The paper is an excellent contribution and should be published, but the nomenclature and discussion is confusing and the authors seem to rely on the reader to be familiar with previous papers to know the various definitions. This should not be assumed. As can be seen from the critique below, some definitions are absent, are made clear only later in the paper or remain confusing. In reading the paper, I remained very confused until near the end of the paper, where plots of data made definitions much clearer. The reader shouldn't have to experience this. A common issue is that when a rheological quantity is defined as a maximum value, it should be explained whether it is a maximum over strain at fixed strain rate or over strain rate after attainment of steady-state or near the point of strand breakage. The theory does not directly predict breakage point, but this is inferred from a Considere criterion, and so the strain at which the stress is evaluated in the theory must be explained clearly. The authors should take on the mindset of an uninformed reader and when a quantity is introduced make sure that it is defined clearly then and there, and clearly enough that a non-specialist reader can clearly understand it. They should introduce symbols that consistently identify when a quantity is a maximum and whether it is maximum over strain or strain rate, or both. For example, using additional subscripts or superscripts on the viscosities to indicate the time or strain at which a quantity is evaluated, such as "ss" for steady-state predictions or "max" for time or strain at maximum stress and using such consistently throughout should help the reader a lot, as explained in more detail below.

Is there a difference between "strain of the maximum stress," "maximum Hencky strain ϵ_q ," and "Hencky strain to failure ϵ_f "? If so, what are the differences? In what sense is "the maximum Hencky strain" a limiting case of "Hencky strain to failure"? In what limit is this true? At high strain rate? Or something else? These quantities have to both be given clear definitions to make this a meaningful statement.

Eq. 7 is valid for small t_q/τ_b , but t_q is the time at the "maximum Hencky strain," and is it clear that this will be small compared to τ_b , especially in the linear viscoelastic regime? Also the term "maximum Hencky strain ϵ_q " at the top of page 3 seems to be the same as the "strain of the

maximum stress" at the bottom of page 2, right? This is confusing, since "maximum strain" suggests a maximum imposed strain, and differs from "strain at maximum stress." The authors should stick to the terminology "strain at maximum stress" or use "epsilon_q" to make the meaning absolutely clear. Actually, the pom-pom model, under homogeneous deformation has a plateau stress at long times, but no overshoot in stress, so how is the "strain at maximum stress" determined for the pom-pom model? Does this actually mean strain required to reach the plateau stress? In the linear regime, the stress maximum is also at steady state, and so t_q should approach infinity. So, how can t_q/τ_b be considered "small"? Or is t_q the time at which the stress is maximum for a high-strain-rate extensional flow? (But even this time is at infinity since stress rises monotonically.) So, t_q should depend on strain rate, and should be a function of strain rate $\dot{\epsilon}$? Then, t_q in Eq. 6 is the value corresponding to some strain rate outside of the linear limit, and it should be explicitly noted that $t_q(\dot{\epsilon})$ is a function of that strain rate.

(If steady-state predictions are taken from the model and used to infer non-steady-state quantities from experiment, such as maximum viscosity before breakage, then notation for model quantities should reflect what is assumed in the model (for example using "ss" for steady-state) should be introduced, and an explanation given for how such model predictions are related to the experimental measurements, and the assumptions required to relate the two should be briefly explained.)

Eq. 9 gives a formula for the "strain at failure" ϵ_f . Again, what is the definition of "strain at failure" ? Presumably it is different from "maximum Hencky strain ϵ_q "? But, we were told that failure occurs, or at least begins at the maximum stress, which occurs at strain ϵ_q . So, why isn't ϵ_f equal to ϵ_q ?

The discussion of "failure mechanism" for linear chains on page 4 may be incomplete. The Considere' criterion only invokes elastic stress dependence on strain and has nothing to do with bond strength. Experimentally, once necking starts, stress in the neck grows uncontrollably, and will eventually reach the bond strength, resulting in bond rupture. But this describes the ultimate fate of the neck, and is not necessarily the initiating "failure mechanism," which is related to the rheological properties, not bond strength. Much later on, we learn that there is a Regime IV, in which the filament evidently fails by bond breakage, while other mechanisms presumably prevail in other regimes, but this is not clear when the breakage condition is first introduced. Instead, page 4 discusses branch point withdrawal and then rupture by bond breakage at high stress, suggesting that this progression occurs as function of time. Much later, we learn that "high stress" is limited to high strain rates in Regime IV, and not the ultimate result as stress increases at lower strain rates. The text early on should be clarified to avoid this confusion. By the way, I believe Suzanne Fielding has corrected the "Considere' criterion" for rupture of viscoelastic liquids, and reference to her work on this topic should be considered.

At the bottom of page 5, the "Considere factor", which seems to be the ratio of extensional viscosity at the stress maximum to linear viscosity at the same time point, is equal to the maximum "strain hardening factor" which is the ratio of extensional viscosity to steady state Doi-Edwards viscosity at the same strain rate, since it is defined on the bottom of page 5 as $\eta_E(\dot{\epsilon})/\eta_{DE}(\dot{\epsilon})$. This suggests that the maximum is with respect to time or strain during start-up of extension. But, in Fig. 5b, the maximum strain hardening factor is a function of strain rate presumably after finding the maximum with respect to time. Thus, the authors need to develop a clearer notation, since SHF_{max} is the maximum with respect to $\dot{\epsilon}$ of the ratio $\eta_E^{max}(\dot{\epsilon})/\eta_{DE}^{max}(\dot{\epsilon})$ where I have here introduced "max" to mean at maximum stress, which would be the peak in the viscosity curve in experiments or steady-state in the theory, right? So, there seem to be two maximizations (with respect to strain and with respect to strain rate) to obtain SHF_{max} . Also confusing is that the y axis in Fig. 5b gives these viscosities as functions of time t and strain rate, but they should be at the point of maximum stress and so do not depend on time. Is this correct? Notation is unclear. A confusion also results from the comparison of the maximum in SHF against f_c , the Considere factor, where the former is taken as a maximum with respect to strain rate and the latter is not. Evidently this is because f_c is roughly independent of strain rate over Regime 3,

and so taking a maximum is unnecessary. But, when these concepts are first introduced, it is very hard to follow what is meant. There is also the question of the experimental definition of these quantities and how they are assessed theoretically, since the pom-pom theory gives extensional viscosities as a function of both extension rate and time.

The following statement on page 7 is confusing: "Due to the unentangled arms, the measured plateau modulus of the sample 280k-2x30-7k is the diluted modulus of the backbone and was adjusted to $G_0 = 8 \times 10^4$ Pa to better fit the experimental LVE." Why is this one polymer singled out for this remark? Is it the diluted modulus that was adjusted to 8×10^4 Pa? This is what the sentence seems to say, but G_0 is supposed to be the undiluted modulus. Perhaps the sentence should say: "Due to the unentangled arms, the measured plateau modulus of the sample 280k-2x30-7k is the diluted modulus of the backbone and to match the experimental LVE the undiluted modulus was adjusted to $G_0 = 8 \times 10^4$ Pa." Also, at this point in the paper the "diluted modulus" is not defined and so its definition should be given here to help the reader understand. If this one sample is the only one for which G_0 was adjusted, is there some explanation for this exception?

In the statement at the bottom of page 10 that "and strain hardening substantially decreases", does "strain hardening" mean the ratio of nonlinear viscosity to linear viscosity at the same point in time? The viscosity at low strain rates looks as steep or steeper at low strains as it is at high rates, and it is only the ratio of viscosity to linear viscosity that decreases. Is this what the term "strain hardening" means? The term "strain hardening" does not seem to be defined in the text, although "strain hardening factor" is defined. Should the reader understand these two terms to be the same thing? If not, then "strain hardening" needs defining.

Presumably the discussion on page 11 extensional viscosity curves "matching" regardless of arm molecular weight presumably means that the curves can be along the time and viscosity axes shifted into overlap, right? This should be stated more precisely than just saying that they "match."

In the caption to Figure 4, "inlet" should be "inset." The inset has no axis labels, and it is hard to know what is plotted. We are told it is the "deviation between the calculated tensile stress growth coefficient and the measured LVE," but does this mean it is the difference $\eta_E^+(t) - \eta_E^+$, LVE (t), between the nonlinear and the linear stress growth coefficients at the time t? This would mean that the curve would tend towards zero at low strain rates. Properly labelling the x and y axes would help the reader understand what is plotted.

On page 12, the statement "overall shape of the tensile stress growth coefficient is similar to the measured tensile stress growth coefficient". Should be "overall shape of the predicted tensile stress growth coefficient is similar to the measured tensile stress growth coefficient."

On page 12, I don't understand what is meant by "the highest measured extensional viscosity is predicted well only from the topology." This seems to mean that only q , the number of arms is required to obtain the highest measured viscosity? While the Considere ratio f_c depends only on q , f_c is the extensional viscosity over its linear viscosity and the latter depends on other factors besides q , doesn't it?

The statement that "the predictions overestimate the LVE between $\sim 1 - 100$ s, by up to 50 %." can't be assessed because the time axis on the insert is not labelled. I don't understand what is meant by "a brief decline in the slope of the measured LVE". Where do I see this reduced slope? Is it reduced with respect to the predicted slope? It is hard to understand what the authors are talking about.

The top of page 13 refers to sample 280k-2x30-7k, saying that the stress growth coefficient for this sample is predicted more accurately than for "280k-2x30-7k and 100k-2x11-9k". But the first of these two samples, 280k-2x30-7k, is the same as the sample which it is said to be more accurate. So, there

is some labeling mix-up here.

The statement "The prediction of the strain rate with the highest measured extensional viscosity is the closest to the measured tensile stress growth coefficient," seems only valid for Fig. 4a, and seems to be referring to the curves for strain rate 0.0056; is this correct? This could be made clearer.

The statement "Equation (13) seems to systematically overestimate the Weissenberg number of the strain hardening onset, suggesting a difference in the relaxation mechanism for combs and pom-poms." To be clear, do you mean that for combs, the strain hardening starts at Wi much above unity, while for these pom-poms, hardening starts at Wi near unity? I don't understand the claim "Stress release of a (partially) stretched backbone can in this picture only be achieved by the reptation of the whole pom-pom molecule." The stretch relaxation time of the pom-pom backbone only requires Rouse motion of the backbone within the tube, which occurs at a rate set by the mobility of the branch points and is not influenced by number of entanglements, while complete orientational relaxation of the backbone requires reptation of the backbone out of the tube. Reptation which should be a factor of Z_{diluted} slower than the Rouse relaxation, where " Z_{diluted} " is the diluted number of tube segments in the backbone. The value of Z_{diluted} for the pom-pom 280k-2x22-22k would seem to be around unity, since it is proportional to the square of the volume fraction of backbone. So, possibly the reason strain hardening occurs at $Wi = 1$, is because the backbone is not self-entangled for this pom-pom. Could similar reasoning be applied to the other pom-poms to see if the onset of strain hardening for them can be explained? In some of them, Z_{diluted} is presumably significantly greater than unity, and they should show strain hardening $Wi > 1$. If this reasoning is correct, then it seems that the analogous combs with same arm and backbone molecular weights, and same number of branches, may require higher Wi to strain harden significantly than for pom-poms because only the most central backbone segment of the comb feels the full stretch possible, since other segments have fewer arms on one side of them and thus do not stretch as much.

These sentences are confusing: "the extensional viscosity is only depending on the backbone length. Together with the Considered limit, these findings suggest that strain hardening of pom-pom systems depends only on the number of arms attached to the branchpoints and not on the length of the arms or the backbone." The first sentence says that the extensional viscosity depends only on backbone length and the second that strain hardening does not depend on backbone length. Again, the term "strain hardening" is not precisely defined, so it is hard to know precisely what is meant. But somehow the definition of strain hardening allows it to not depend on backbone length although the viscosity does depend on it? This is confusing.

The scaling law of extensional viscosity with -0.5 exponent is not explained, and yet is crucial to the prediction of the Considered criterion. If the pom-pom model is used, the linear rheology does not show a -0.5 power law, and this must arise from dynamics not in the pom-pom model. But the derivation of the Considered factor after Eq. 11 requires using the pom-pom linear rheology formula Eq. 6, which does not follow a power law of -0.5 , and substituting its low-time expansion into Eq. 5. But the Considered' criterion is related to the nonlinear rheology, not the linear rheology and so this substitution seems to be cancelling out an error in the prediction of the nonlinear behavior by a proportional error in the linear rheology. Why should we consider the Considered criterion expressed as a ratio to a linear response the most valid expression? The authors should discuss this and at least note that the expression for the linear behavior is not accurate and this has implications for the validity of Eq. 11 for their data.

Are the orange data points given in Fig. 6 experimental data points? If so, the sample should be named in the caption. If not, please note in the caption that they are schematic.

The summary discusses Considered factor and maximum strain hardening factor, but these should be defined clearly also in the summary so that the reader who skims the article or forgets the definitions is refreshed.

Minor points:

Since I am not an expert on organic synthetic chemistry, I confine my comments to the rheological portion of the paper.

In Eq. 1, the stress tensor σ should be bold since it is a tensor.

Response to the Reviewers

We very much thank the Reviewers for their interest in our manuscript and for their thoughtful and constructive feedback, which has helped us to improve our manuscript. In the following, we respond point by point to the comments.

Reviewer #1 (Remarks to the Author):

This work analyzes the extensional rheology for a series of Pom-Pom polystyrenes with well-defined molecular architecture, and links the rheological behavior especially the maximum strain hardening factor to the molecular topology. Extensional rheology data of model branched polymer is rare. This work provides valuable experimental data that improves our understanding of polymer dynamics in strong flow. In particular, the generalized relationship between the strain hardening and the molecular topology of Pom-Poms in Fig.6 of the paper is very interesting and provides an important guidance for molecular design of branched polymers. However, there are also a few points requiring further clarification from the authors before the work can be published:

1. *The samples are all measured at the same reference temperature, but do they have the same glass transition temperature T_g ? Do the short arms influence T_g ?*

Response: Thank you for this important comment. The glass transition temperature for all samples is around 105 °C as expected for high molecular weight polystyrene. Four examples are shown in Figure A1, which clearly have T_g around 105 °C.¹ Similar results have been obtained for example by *van Ruymbeke et al.*¹ with even shorter arms ($M_{w,a} = 4.4 \text{ kg mol}^{-1}$, $M_{w,b} = 36 - 182 \text{ kg mol}^{-1}$). This suggests that short arms (still far above the Kuhn length) on long backbones do not significantly decrease the T_g .

Figure A1: Heat flow curves of four pom-poms samples at a heating rate of 10 K min⁻¹. Second heating runs are shown.

2. *In Fig.3a, the sample 100k-2x12-24k shows only one rubber plateau, and the authors relates this plateau to entangled arms and concludes that the backbone is not entangled. But I think*

this plateau should be due to entanglements including arm-arm, arm-backbone, and backbone-backbone interactions. How do the authors know this plateau is from arm-arm entanglements only? Furthermore, in Fig.3b, what does the low-frequency minimum of 100k-2x12-24k mean (although $G'' > G'$)?

Response: Thank you for pointing this out. The rubber plateau is, within the uncertainty of the experiment, the same for all samples with the same number and molecular weight of the arms and irrespective of the backbone molecular weight. We think that the high frequency rubber plateau of sample 100k-2x12-24k is therefore corresponding to arm-arm and arm-backbone entanglements. For samples with self-entangled backbones, i.e., two rubber plateaus, the high frequency rubber plateau is attributed to arm-arm, arm-backbone, and backbone-backbone interactions. The low frequency rubber plateau then results from backbone-backbone entanglements since the arms are already relaxed.

We attribute the low-frequency minimum to the backbone and therefore backbone-backbone entanglements, if the arms do not dilute the backbone above a certain degree, thus only Rouse like behaviour can be observed. More backbone-backbone entanglements would lead to a lower phase angle in the low ω minimum of the mastercurve due to more elastic behaviour from to the higher number of entanglements. This is supported by the correlation of the depth of the low-frequency δ -minimum with the volume fraction of the backbone as shown in Figure A2.

Figure A2: a) Van Gurp-Palmen plot of the pom-pom 400k-2x12-23k. b) Minimum of the phase angle at low complex moduli as a function of the backbone volume fraction, showing for the 25 and the 40 kg mol^{-1} series a good correlation.

3. In Fig.4a, the predictions of the pom-pom model underestimate the steady state viscosity severely especially at fast rates. However, I am wondering if the authors checked the real Hencky strain of the measured samples. For example, the simulations from *J. Non-Newtonian Fluid Mech.* 165, 14–23 (2010) show that for highly elastic materials stretching on SER, the actual Hencky strain can deviate from the ideal Hencky strain significantly (also depends on initial sample size). When the ideal Hencky strain is 2, the actual Hencky strain of the sample can be as low as less than 1. That means the actual cross-section area is much larger, and the corresponding stress should be lower. Stretching samples on EVF may have the same problem, especially for the samples with high strain hardening. Therefore, I think the authors should check the cross-section area at the end of stretching for at least one sample (which shows high strain hardening and deviates significantly from pom-pom model prediction).

Response: Thank you for this very thoughtful comment. We agree that the SER and the EVF geometry are indeed very similar and therefore it is reasonable to assume the calculation can be applied to both geometries. To assure ideal deformation, the mentioned reference recommends, that the sample ratio of width to length should be below 0.2. From an experimental point, we could not observe any influence of the cross-section area on the measured stress using the EVF geometry on a strain-controlled ARES-G2. While varying the width between 2 – 10 mm and the thickness between 0.2 – 1 mm, we could not observe any influence on the measured stress, as shown for some examples in Figure A3. The extensional measurements in the manuscript are conducted with specimen of the dimensions 15mm*5mm*0.5mm, i.e. the width to length ratio was 0.33.

Figure A3: Time dependent tensile stress growth coefficient for the pom-pom 280k-2x22-22k at 150°C and a Hencky strain rate of 1 s⁻¹ and different specimen width and thickness. The results are within experimental error identical, indicating (within this range) no influence of the specimen geometry.

4. In Eq.4, it would be helpful if the authors can further clarify the difference between “the Hencky strain-to-failure” and “the maximum Hencky strain” – why the Hencky strain-to-failure is not the “maximum” Hencky strain?

Response: Thank you for pointing out this unclear definition of quantities. The Hencky strain to failure ϵ_f is the strain at which the sample fails in the experiment. The maximum Hencky strain ϵ_q is the Hencky strain at which the backbone reaches its maximum stretch $\lambda = q$ in the pom-pom model. This is the case in regime III where then the Considère limit applies. In e.g., regime II the maximum backbone stretch is not reached and therefore no branch point withdrawal is taking place.

We revised the introduction and added Figure 1a) to make the definition of each quantity more precise and easier to understand:

The pom-pom model attributes two relaxation times to the backbone. The orientation relaxation times τ_b , describing the alignment of the backbone tube due to flow, and the stretch relaxation time τ_s , describing the stretch of the backbone chain in flow. In extension with $\dot{\epsilon} > 1/\tau_s$, the backbone is stretched due to the friction of the branchpoints. At sufficiently high strains, the backbone reaches its maximum stretch $\lambda = q$ and thereafter the arms are retracted into the

backbone tube, resulting in a steady state stress. The retraction of the arms is called branch point withdrawal and takes place at the time t_q and strain ε_q of maximum backbone stretch. Hierarchical relaxation of the arms before the backbone is essential in the pom-pom model.

5. From Eq.11, f_c is defined as the ratio of the extensional viscosity and the corresponding startup viscosity on the LVE; but on page 5 line 119, f_c is equal to the ratio of the extensional viscosity and the steady state viscosity of the Doi-Edwards model. This is a little bit confusing - are the two definitions equivalent to each other?

Response: Thank you for pointing this out. f_c and SHF_{max} indeed have slightly different definitions against the LVE and the steady state viscosity of the Doi-Edwards model, respectively. The finding that they are equal for our pom-pom samples within a narrow strain rate window is of experimental nature. We added to the manuscript in line 422 to clarify the experimental finding: “Note that the Considère factor is normalized against the LVE $f_c = \frac{\eta_E(\dot{\varepsilon})}{\eta_{LVE}^+(1/t)} \simeq \left[\frac{q^2}{\ln(\sqrt{3}q)} \right]$. When comparing the plateau viscosity and the steady state viscosity of the DE model, as shown in Supplementary Figure 17, one can observe $\eta_{LVE}^+(1/t) = \eta_{DE}(\dot{\varepsilon})$ within the uncertainty of the experiment. $SHF_{max} = f_c$ can therefore directly be derived from their definitions above.”

6. On page 13 line 270, the authors mentioned that “ τ_l is the longest relaxation time of a sample, determined by the crossover into the terminal regime”. Is τ_l of the 100k backbone sample also determined the same way? From Fig.3a, the crossover into the terminal regime of 100k-2x12-24k is apparently not corresponding to the longest relaxation time.

Response: Thank you for pointing out this missing information. We added to the manuscript in line 297: “The longest relaxation time of the samples with not self-entangled backbones, namely 100k-2x12-24k and 100k-2x12-40k is obtained by fitting a single mode Maxwell model to the terminal regime of G' and taking the crossover frequency of the Maxwell model as τ_l .”

7. On page 16, line 334-337, the authors mentioned that “the extensional viscosity is only depending on the backbone length” and “strain hardening depends not on the length of the backbone.” It is not very straightforward to understand and would be helpful if the authors can explain it further.

Response: Thank you for pointing towards this unclear statement. We clarified in the manuscript from line 367 on: “As shown in Supplementary Figure 2 and discussed earlier, the extensional viscosity is only depending on the backbone length and the arm number, independent of $M_{w,a}$. Together with the Considère limit, these findings suggest that strain hardening (factor) ($SHF = \eta_E(\dot{\varepsilon})/\eta_{DE}(\dot{\varepsilon})$) of pom-pom systems depends only on the number of arms attached to the branchpoints and not on the length of the arms or the backbone. This SHF is given by the Considère limit as a function of the arm number.”

8. In Fig.7b, would it be better to use the extensional stress (instead of the extensional viscosity) to normalize with the plateau modulus?

Response: Thank you for your suggestion. The extensional stress normalized to the plateau modulus is shown in Figure A4. The experimental data shows the same power law as in Figure 7 in the manuscript and does not show any additional information compared to the extensional viscosity.

Figure A4: Extensional stress normalized to the plateau modulus as a function of the volume fraction of the backbone.

Reviewer #2 (Remarks to the Author):

Here the authors study linear rheology and extensional flow behaviour of pom-pom type polymers, synthesised to have a large number of arms at the end of the polymer. They analyse the behaviour in terms of the pom-pom model of McLeish and Larson, making use of a Considere criterion for the elongation at break. Unfortunately a lot of the analysis is performed in a way that betrays a misunderstanding of the pom-pom model and the expected molecular relaxation process, and the limitations of the model. So, I think quite a lot of the analysis is flawed or incomplete. I'll try to work through the paper detailing places where more consideration is really needed.

Response: We thank the reviewer for the extensive review of our manuscript. We hope to correct and clarify all the raised issues.

1) In the introduction, leading to Eqs 1-11, more detail is needed, especially on what assumptions are made along the way and what limits are being taken. For example, assumptions are made as to strain rate and strain in order to find eq 3 as a limiting case from eq 1. For eq 4 it is not really explained what the difference is between strain to failure and maximum strain; again in eq 9 there are assumptions made about the stretching rates. These all need detailing otherwise the reader needs to look back at the original papers and figure it out for themselves.

Response: Thank you for pointing out the missing assumptions. We thoroughly revised the manuscript to include all assumptions to make it more accessible without looking back at the original papers.

2) Also eq 12 is not clear... the right hand side is given in terms of angular frequency (ω) which seems odd in context. Surely we want a function of time?

Response: Thank you for pointing this out. We corrected the equation and added to the manuscript in line 108: “...taking the Trouton ration, the Cox-Merz and Gleissle mirror rule into account.”

3) In the methods section on rheology, it should be discussed what the maximal extensional strain of the instrument is, and whether it is ever reached before breaking (or does the sample always break first?)

Response: Thank you for this suggestion. The limitations of the EVF geometry are also discussed in the paragraph starting line 250. For further clarification, the following section has been added to the manuscript methods section in line 182: “The measurement of the tensile stress growth coefficient is limited to a Hencky strain of $\varepsilon \leq 4$. Generally, for $q > 22$, at high rates the sample fails at $\varepsilon < 4$, at medium strain and low rates the sample does not fail beforehand. However, for all samples the stress maximum is reached for $\varepsilon < 4$. For $q \leq 13$, samples fail $\varepsilon < 4$.” This is also shown exemplary in Figure A as a function of time to compare against the elongational data of Figure 3 or Supplementary Figure 3.

Figure A5: Maximum Hencky strain shown as a function of time for three pom-pom samples. Lines are guide to the eye.

To evaluate if it is justified to assume that $\eta_E^+(\varepsilon = 4) = \eta_{E,steady-state}^+$ for the measurements where the sample does not fail before $\varepsilon = 4$, we show unpublished measurements for highly branched combs in Figure A. Molecular parameters of the combs are similar to the investigated pom-poms. The measurement of the extensional stress on the EVF geometry (limit $\varepsilon = 4$) is compared to measurement on a filament stretch rheometer (FSR). Although the sample can be

stretched further, the steady state stress is reached around $\varepsilon \sim 4$. Judging from this, we think that the above assumption is justified.

Figure A6: Extensional stress as a function of Hencky strain for a highly branched comb. Measurements were obtained on an EVF geometry (red triangles) and on a filament stretch rheometer (black squares).

4) Figure 3c (and other similar figures in the main body of the paper) only really shows the highest extension rates (I think typically regions III and IV from the later analysis), with the full range of extension rates only in the supplemental info. I think it's important to see at least one instance of the full range of rates.

Response: We welcome your comment and added a fourth graph to Figure 3 in the manuscript showing the extensional viscosity of the pom-pom 400k-2x12-23k for all measured strain rates.

5) Figure 4: it's important to detail what parameters were used for the pom-pom predictions, especially what values of orientational and stretch relaxation times. The orientation time should be close to the longest relaxation time. Looking at the extended predictions in SI, it seems that often the LVE at long times is underpredicted by the parameters used, ... but this is where the predictions should be matched because the long times correspond to the backbone relaxation. Matching at intermediate times around $t=10$ corresponds to angular frequencies of ~ 0.1 in the LVE, which is no-where near the backbone orientation time. Hence I think the pom-pom fitting may not be done correctly.

Response: Thank you for your suggestions towards the fitting of the pom-pom model. The parameters used are the following: s_a , s_b , q , G_0 , $\tau_{R,e}$, concentration c , dilution exponent α , $\dot{\varepsilon}$, duration of elongation t . All other parameters are calculated based on these inputs described in McLeish, T. C. B., et al. *Macromolecules* 32, 6734-6758, (1999) using the IRIS Rheo-Hub software. In Figure A below, all modelling is done using $\tau_e = 0.036$ s, $G_0 = 2 \cdot 10^5$ Pa, $c = 1$, $\alpha = 1$ and s_a , s_b , q as stated in table 1. We chose these parameters as they are typical material values for polystyrene or molecular parameters for the samples determined from the synthesis. Clearly, the pom-pom model does not really match the experimental LVE. As a result, the extensional data is also not described well. Whereas, the strain hardening, so the increase of the extensional viscosity above the LVE, is roughly correct.

Figure A7: Tensile stress growth coefficient η_E^+ as a function of time for different pom-pom samples at a reference temperature of $T_{ref} = 140$ °C. Dots show experimental data, solid lines show simulated tensile stress growth coefficient calculated by the pom-pom model with the respective molecular parameters and PS specific material parameters.

Figure A7 continued: Tensile stress growth coefficient η_E^+ as a function of time for different pom-pom samples at a reference temperature of $T_{ref} = 140$ °C. Dots show experimental data, solid lines show tensile stress growth coefficient calculated by the pom-pom model with the respective molecular and PS specific material parameters.

McLeish and Larson mention in J. Rheol. 1998, 42, 81 that neither the backbone nor the arm dynamics should dominate the pom-poms response and therefore they calculated a pom-pom with $s_a = 3$, $s_b = 30$ and $q = 5$ resulting in a backbone and arm volume fraction of $\phi_b = \phi_a = 0.5$. Due to the high arm numbers in our samples, all ϕ_b are below 0.5 with the 400k-2x12-23k sample at $\phi_b = 0.42$ having the highest. If the plateau modulus is raised to $G_0 = 4 \cdot 10^5$ Pa, a good fit between the experimental extensional data of 400k-2x12-23k and the pom-pom model can be achieved. Nevertheless, the plateau viscosity is underpredicted by factor of ~ 10 , revealing different backbone relaxation times. The LVE can be represented by the pom-pom model if $G_0 = 1.8 \cdot 10^5$ Pa, $\tau_e = 0.05$ s and $s_b = 38$ are chosen and therefore the backbone relaxation times are artificially increased.

Figure A8: Tensile stress growth coefficient η_E^+ as a function of time for different pom-pom samples at a reference temperature of $T_{ref} = 140$ °C. Dots show experimental data, solid lines show simulated tensile stress growth coefficient calculated by the pom-pom model with the respective molecular parameters, PS specific material parameters but increased modulus G_0 .

The Chapter Constitutive Modelling was revised accordingly.

6) page 13, There is a confusing sentence: "For the sample 280k-2x30-7k, the experimental and calculated tensile stress growth coefficients are shown in Figure 4b). Overall, the tensile stress growth coefficient is predicted more accurately than for samples with unentangled arms (280k-2x30-7k...." The two occurrences of 280k-2x30-7k don't make sense together!

Response: Thank you. We revised the chapter Constitutive Modelling, and both sentences have been removed.

7) Page 13: it is worth noting that the crossover time in LVE is not always the best measure of longest relaxation time, especially when a lot of dynamic dilution has taken place. Likely the longest relaxation time is a bit longer than estimated here.

Response: Thank you for raising this important issue. We thoroughly considered how the longest relaxation time could be determined in a precise and reliable way. Due to different approaches to obtain the longest relaxation time, we chose the crossover frequency to be reliable and can be determined very precisely for many samples. From the subsequent analysis, we now find that the inverse crossover frequency seems to be roughly similar to the stretch relaxation time of the backbone. While the orientation time of the backbone being slightly longer, see discussion of comment 8) below.

8) Page 14: "As $Wi, De > 1$, the backbone chains are not fully relaxed, and the backbone polymer chain can be partially both oriented and stretched during elongation...." and likewise page 15: "terminal relaxation time of pom-poms is equal to the experimentally observed onset of strain hardening." and "Stress release of a (partially) stretched backbone can in this picture only be

achieved by the reptation of the whole pom-pom molecule." Here the authors have not understood the difference between the orientation relaxation time and stretch relaxation time in the pom-pom model. Typically the orientational relaxation time is the longest relaxation time in LVE, and for flow rates greater than inverse of orientation time some extension thinning is expected because tubes orient but chains do not stretch. This is seen to the left of the $Wi=1$ line in fig 5 (an indication that the longest relaxation time is a little underestimated). Once the flow rate is greater than the inverse of the stretch time then chains stretch and extension hardening is seen. For backbones with only a few (diluted) entanglements, as in 280k-2x22-22k, the stretch and orientation times are quite close, but they are not the same thing.

Response: Thank you for your detailed insights. We are aware that orientation is a slow process compared to stretch and corrected our statements as follows:

From line 313:

In this regime, the backbone as well as the side chains are in their equilibrium stretch, and the extensional viscosity matches the LVE. Between $\sim 0.3 < Wi < 1$, shear and extensional thinning can be observed, indicated by the decrease in the normalized viscosity, caused by orientation of backbone chains due to the flow faster than the orientation relaxation time τ_b . However, since the flow is slower than the stretch relaxation time τ_s , no strain hardening can be observed here.

From line 320:

As $Wi, De > 1$, the backbone chains are stretched out of their equilibrium stretch due to $\dot{\epsilon}^{-1} > \tau_s$. Therefore, an increase of the extensional viscosity above the LVE can be found, thus strain hardening is observed.

From line 324:

Similar to a ring-shaped sample, the crossover relaxation time of pom-poms seems roughly similar to the experimentally observed onset of strain hardening and therefore the stretch relaxation time. The thinning observed in regime I indicates a longer orientation relaxation time.

From line 343:

Stress release of a (partially) stretched backbone can in this picture only be achieved by movement of the whole branch point. This molecular hypothesis could explain the experimentally observed onset of strain hardening of the pom-poms at the inverse of the crossover frequency.

9) Eq 9: it is a good idea to estimate the Rouse time as with equation 13, which is for combs. Likely the Rouse time of a pom-pom will be a bit larger than this, because for a comb the monomer friction from arms is distributed along the backbone, whilst for the pom-pom it's all at the chain ends. There is some confusion about τ_e ... it's given as 0.036s on page 15 and 0.225s on page 7. I think the 0.036s is probably closer, but I'd estimate a bit bigger than that. In any case, for flow rates above the inverse of the Rouse time, the pom-pom model no longer applies because at these rates the monomer friction alone is sufficient to stretch the chains and branch-point withdrawal (which happens at the rate of the Rouse time) can no longer occur. The pom-pom model assumes flow rates smaller than inverse of Rouse time. Here, then, we expect chains to continue stretching until maximal extension.

Unfortunately this means that the pom-pom model should not be applied at all at these high rates, which corresponds to regimes III and IV. So, in these regions, the Considère criterion derived from the pom-pom model should not apply.

I suspect that what is happening in practice here is that there is a combination of chains reaching their maximal stretch through finite extensibility and of the reduction of chain friction (as seen in oriented linear chains). This gives rise to a -0.5 slope for strongly stretched linear polystyrene chains, and I think it's the same thing happening here.

Response: Thank you again for your insights and extensive review.

The two different τ_e are due to the fitting of the pom-pom model (0.225s) to the experimental data on the one hand. On the other hand, 0.036s is used as recommended in Lentzakis et al. for Eq. 13. We adjusted the modelling based on an earlier comment.

Thank you for pointing out the limit of the pom-pom model. As you estimated, $\tau_{R,eq.13}$ should be slightly smaller than $\tau_{R,pom-pom}$. In a first approximation and since we don't have a more sophisticated equation we neglect the different mass distributions along the backbone, and therefore assume $\tau_{R,eq.13} = \tau_{R,pom-pom}$. The strain rate with the highest overall extensional viscosity is then for all samples within the limits of the model, $\dot{\epsilon} > 1/\tau_{R,pom-pom}$. Hence the pom-pom model and the Considère limit, can be applied to predict the highest extensional viscosity successfully and within the model assumptions. The wide range of arm numbers ($q = 9 - 22$) investigated, affirms this finding of the Considère Limit.

By contrast, we surely agree that it is very surprising that the Considère limit can apparently predict the experimental data very well within the model limits and even outside the validity beyond τ_R down to $\tau_{\dot{\epsilon},frac}$. This needs to be investigated in detail in the future, for which this data set hopefully can provide aid.

Thank you again for your insights, we adjusted all "Considère graphs" accordingly.

10) Page 15: "Equation (13) seems to systematically overestimate the Weissenberg number of the strain hardening onset, suggesting a difference in the relaxation mechanism for combs and pom-poms."

This sentence betrays a fundamental misunderstanding of two different stretch relaxation times in branched polymers (pom-poms or combs). The "stretch relaxation time" in the pom-pom model is derived by considering that the branch-points take hops at the timescale of entangled arm relaxation. This hopping gives rise to an effective friction from the hopping motion, and the stretch relaxation time is found by balancing the hopping friction with the backbone spring constant. If the flow rates are faster than this stretch time, the pom-pom model predicts the backbone stretches up to the maximum stretch given by the priority, at which point branchpoint withdrawal occurs. There is a directly analogous stretch relaxation time for comb polymers (again from hopping of the branchpoints, and again with the possibility of branchpoint withdrawal) as described in ref 44. The Rouse time, on the other hand, is just the stretch relaxation time for the backbone pulling against MONOMER friction (not branchpoint hopping friction). As detailed above, if the flow is faster than the Rouse time then chains stretch and do not even have time for branchpoint withdrawal. Clearly the Rouse time is shorter than the pom-pom stretch relaxation time.

Response: We gratefully take your comment into account and removed the statement. Thank you.

11) For the reasons given in point 9 and 10 above, the main conclusion for region III is wrong, e.g as described in page 15, line 321: "At higher Weissenberg numbers, regime III can be found between the phase angle minimum of the backbone and the relaxation time of the arms, $Wi_{\delta,b,min} < Wi < Wi_{\tau,a}$. In this regime, we experimentally find that the extensional viscosity (I) declines with a scaling exponent of about -0.5 and (II) matches the prediction of the Considère limit, a factor of $[q^2/\ln(\sqrt[3]{3}q)]$ above the LVE as stated in equation (11)."

Simply put, you can't use the Considère criterion derived from the pom-pom model at flow rates outside the regime of validity of the pom-pom model.

Response: Thank you again for pointing out the limitations of the model. As stated in the response to point 9, we think that there is a range of strain rates, where the pom-pom model and the Considère Limit can be successfully applied within the model limits, especially at the strain rate with the highest extensional viscosity. Also, within our article, we point out the limit of strain rates where the Considère limit can be applied.

Additionally, we describe experimental observations made for our samples and to emphasize this, we added the following to the manuscript in line 353: "These experimental findings need to be considered carefully, since the pom-pom model is only valid for $\dot{\epsilon} > 1/\tau_R$, which is estimated to be in the lower Wi range of regime III. For the extension rate at the η_E maximum at $Wi_{\delta,b,min}$, the strain rate is within the limits of the pom-pom model and its assumptions are justified. However, experimentally we find that the prediction from the Considère criterium is a good approximation of η_E within the whole regime III."

12) Finally, to regime IV, described from page 16 onwards. Here we are at rates much faster than the Rouse time... indeed the rates are approaching the entanglement time of PS. This happens coincidentally to be a similar order of magnitude to the arm relaxation time, here because the arms are quite short, around an entanglement length long. So, really the physics here is that ALL the molecule is stretching on scales around an entanglement length, arms, backbone, everything...there is not even time to redistribute monomers along the backbone tube. At these rates it is no surprise that chains might get stretched enough to break, although I suspect the mode of breaking is more subtle than described in ref 17.

Response: Thank you for raising this interesting point, which highly helps to further improve the manuscript. After careful reconsideration of our experimental data, we have changed our interpretation. As you assessed, the arm relaxation time of the pom-pom model systems investigated and an entanglement relaxation time is in the same order of magnitude, but the entanglement time of a segment seems to be smaller than the onset of the plateau of the fracture stress. In Figure A9 a) - d), the extensional stress is shown as a function of the strain rate. Typically, the highest investigated strain rate is $\dot{\epsilon} = 10 \text{ s}^{-1}$ at $T_{ref} = 140 \text{ °C}$. $\tau_e = 0.036 \text{ s}$ translates to a strain rate of $\dot{\epsilon} \sim 27 \text{ s}^{-1}$ at $T = 140 \text{ °C}$. Especially in Figure A9 a), b) and d), the onset of the plateau is much earlier than $\dot{\epsilon} \sim 27 \text{ s}^{-1}$. Additionally, the transition from regime IV

(no slope) to regime III (slope 0.5) is clearly a function of the arm length and the backbone length (Figure A9 c)).

Figure A9: a) -d) Fracture stress shown as a function of strain rate for four pom-pom series of different arm and backbone lengths. All strain rates are shown at a reference temperature of $T_{ref} = 140^\circ C$. e) Onset strain rate of the plateau stress shown as a function of span molecular weight for pom-poms with $q = 12 \pm 1$.

We evaluated the influence of the span molecular weight $M_{w,span} = M_{w,b} + 2M_{w,a}$ on the onset strain rate of the plateau. The onset fracture strain rate as a function of $M_{w,span}$ is shown in

Figure A9 e). A power law of -2.4 can be identified for the onset strain rate. Therefore, we conclude, that the onset fracture strain rate depends on the span molecular weight and the arm number.

We revised the manuscript according to the new insights starting from line 376 on.

Overall: I think there are serious misunderstandings of the molecular physics and of the pom-pom model displayed in the analysis. I think this means the conclusions are quite flawed.

Having said all that: the data is really good and valuable. There are measurements of these samples across a broad range of extension rates and that would be an excellent resource to the community.

Response: Thank you for your positive statement about our work, your extensive revision, and deep insights into the pom-pom model, which helped to improve the manuscript and further strengthen the conclusions. We hope we could improve our manuscript sufficiently and the data can be of use to many other scientists.

Reviewer #3 (Remarks to the Author):

The paper is an excellent contribution and should be published, but the nomenclature and discussion is confusing and the authors seem to rely on the reader to be familiar with previous papers to know the various definitions. This should not be assumed. As can be seen from the critique below, some definitions are absent, are made clear only later in the paper or remain confusing. In reading the paper, I remained very confused until near the end of the paper, where plots of data made definitions much clearer. The reader shouldn't have to experience this. A common issue is that when a rheological quantity is defined as a maximum value, it should be explained whether it is a maximum over strain at fixed strain rate or over strain rate after attainment of steady-state or near the point of strand breakage. The theory does not directly predict breakage point, but this is inferred from a Considere criterion, and so the strain at which the stress is evaluated in the theory must be explained clearly. The authors should take on the mindset of an uninformed reader and when a quantity is introduced make sure that it is defined clearly then and there, and clearly enough that a non-specialist reader can clearly understand it. They should introduce symbols that consistently identify when a quantity is a maximum and whether it is maximum over strain or strain rate, or both. For example, using additional subscripts or superscripts on the viscosities to indicate the time or strain at which a quantity is evaluated, such as "ss" for steady-state predictions or "max" for time or strain at maximum stress and using such consistently throughout should help the reader a lot, as explained in more detail below.

Response: We thank the respected Reviewer for their positive statement about our paper. We thoroughly revised our manuscript following your suggestions for increasing the clarity.

Is there a difference between "strain of the maximum stress," "maximum Hencky strain ϵ_q ," and "Hencky strain to failure ϵ_f "? If so, what are the differences? In what sense is "the maximum Hencky strain" a limiting case of "Hencky strain to failure"? In what

limit is this true? At high strain rate? Or something else? These quantities have to both be given clear definitions to make this a meaningful statement.

Response: Thank you for pointing out this unclear definition of quantities. As also discussed in Comment 4 of Reviewer 1, the Hencky strain to failure ϵ_f is the strain at which the sample fails in the experiment. The maximum Hencky strain ϵ_q is the Hencky strain at which the backbone reaches its maximum stretch $\lambda = q$. We revised the introduction of our article and added Figure 1a) to make the definition of each quantity more precise and easier to understand.

Eq. 7 is valid for small t_q/τ_b , but t_q is the time at the “maximum Hencky strain,” and is it clear that this will be small compared to τ_b , especially in the linear viscoelastic regime?

Response: t_q is the time at which the stress reaches the steady state value in the pom-pom model. In the manuscript in Figure 4b), tensile stress growth coefficient calculated by the pom-pom model are shown up to a Hencky strain of 6. The steady state value is reached within at all strain rates. τ_b can be estimated at $7 \cdot 10^4$ s at $T_{ref} = 140$ °C in this graph from the experimental LVE by the onset of shear thinning. All t_q are smaller compared to τ_b . In addition, the Considère limit in this manuscript is only relevant for strain rates at which strain hardening occurs. Since strain hardening is limited to strain rates that are faster than the backbone stretch time, all t_q are small compared to τ_b .

Also the term “maximum Hencky strain epsilon_q” at the top of page 3 seems to be the same as the “strain of the maximum stress” at the bottom of page 2, right? This is confusing, since “maximum strain” suggests a maximum imposed strain, and differs from “strain at maximum stress.” The authors should stick to the terminology “strain at maximum stress” or use “epsilon_q” to make the meaning absolutely clear.

Response: Thank you for raising this issue. We adjusted the manuscript for better description of the quantities. Please also see Reviewer 1, comment 4:

Thank you for pointing out this unclear definition of quantities. The Hencky strain to failure ϵ_f is the strain at which the sample fails in the experiment. The maximum Hencky strain ϵ_q is the Hencky strain at which the backbone reaches its maximum stretch $\lambda = q$. This is the case in regime III where then the Considère limit applies. In e.g., regime II the maximum backbone stretch is not reached and therefore no branch point withdrawal is taking place.

We revised the introduction to make the definition of each quantity more precise and easier to understand from line 50 on:

The pom-pom model attributes two relaxation times to the backbone. The orientation relaxation times τ_b , describing the alignment of the backbone tube due to flow, and the stretch relaxation time τ_s , describing the stretch of the backbone chain in flow. In extension $\dot{\epsilon} > 1/\tau_s$, the backbone is stretched due to the friction of the branchpoints. At sufficiently high strains, the backbone reaches its maximum stretch $\lambda = q$ and thereafter the arms are retracted into the backbone tube. The retraction of the arms is called branch point withdrawal and takes place at the time t_q and strain ϵ_q of maximum backbone stretch. Hierarchical relaxation of the arms before the backbone is essential in the pom-pom model.

Actually, the pom-pom model, under homogeneous deformation has a plateau stress at long times, but no overshoot in stress, so how is the “strain at maximum stress” determined for the pom-pom model? Does this actually mean strain required to reach the plateau stress? In the linear regime, the stress maximum is also at steady state, and so t_q should approach infinity. So, how can t_q/τ_b be considered “small”? Or is t_q the time at which the stress is maximum for a high-strain-rate extensional flow? (But even this time is at infinity since stress rises monotonically.) So, t_q should depend on strain rate, and should be a function of strain rate $\dot{\epsilon}$? Then, t_q in Eq. 6 is the value corresponding to some strain rate outside of the linear limit, and it should be explicitly noted that $t_q(\dot{\epsilon})$ is a function of that strain rate. (If steady-state predictions are taken from the model and used to infer non-steady-state quantities from experiment, such as maximum viscosity before breakage, then notation for model quantities should reflect what is assumed in the model (for example using “ss” for steady-state) should be introduced, and an explanation given for how such model predictions are related to the experimental measurements, and the assumptions required to relate the two should be briefly explained.)

Response: Thank you for pointing towards this limit. We revised the introduction to give more precise definitions of the quantities. t_q is the time at which the strain on the backbone causes the stars (poms) at each end to withdraw into the backbone tube, or branchpoint withdrawal. For $t \geq t_q$, the stress of the pom-pom model is constant at its plateau value. The strain at maximum stress ϵ_q is determined when the plateau stress is reached at t_q . As discussed in an earlier comment, τ_b is indicated by the onset of shear thinning. τ_b can for example be found around $7 \cdot 10^4$ s for pom-pom 400k-2x12-23k in the experimental LVE and therefore all experimental extensional data is multiple orders of magnitude smaller. In addition, Figure A10 shows the LVE calculated by the exponential Equation 6 and the linear approximation from Equation 7. By comparison of the two LVEs, we can also see the validity of the assumption $t_q \ll \tau_b$ for our experimental data, especially for strain rates of $\dot{\epsilon} = 0.1 \text{ s}^{-1}$ and higher where the Considère limit is applied.

The strain at maximum stress is determined for the pom-pom model at the onset of the steady-state value. Similarly, the extensional viscosity predicted by the Doi-Edwards model is obtained.

Figure A10: Extensional viscosity at different strain rates shown for the pom-pom 400k-2x12-23k at a reference temperature of $T_{ref} = 140$ °C. Experimental and calculated LVEs (Eq.6 pom-pom model, Eq. 7 linear approximation) are shown for comparison.

Eq. 9 gives a formula for the “strain at failure” ϵ_f . Again, what is the definition of “strain at failure”? Presumably it is different from “maximum Hencky strain ϵ_q ”? But, we were told that failure occurs, or at least begins at the maximum stress, which occurs at strain ϵ_q . So, why isn't ϵ_f equal to ϵ_q ?

Response: Thank you for your comment. The Considère criterium assumes, that until the strain at maximum stress ϵ_q , the deformation can be considered homogenous. After the maximum stress the sample is assumed to start necking, ultimately resulting in failure of the specimen at ϵ_f . We revised the introduction and added Figure 1a) to the manuscript to give more precise definitions of the quantities.

The discussion of “failure mechanism” for linear chains on page 4 may be incomplete. The Considère' criterion only invokes elastic stress dependence on strain and has nothing to do with bond strength. Experimentally, once necking starts, stress in the neck grows uncontrollably, and will eventually reach the bond strength, resulting in bond rupture. But this describes the ultimate fate of the neck, and is not necessarily the initiating “failure mechanism,” which is related to the rheological properties, not bond strength. Much later on, we learn that there is a Regime IV, in which the filament evidently fails by bond breakage, while other mechanisms presumably prevail in other regimes, but this is not clear when the breakage condition is first introduced. Instead, page 4 discusses branch point withdrawal and then rupture by bond breakage at high stress, suggesting that this progression occurs as function of time. Much later, we learn that “high stress” is limited to high strain rates in Regime IV, and not the ultimate result as stress increases at lower strain rates. The text early on should be clarified to avoid this confusion. By the way, I believe Suzanne Fielding has corrected the “Considère' criterion” for rupture of viscoelastic liquids, and reference to her work on this topic should be considered.

Response: We thank the reviewer raising attention for the incomplete discussion. We clarified our statements regarding at which strain rates which failure mechanisms occur.

Thank you for bringing the work of Suzanne Fielding to our attention and the limitation for rapid stretching towards the Considère Criterium. We added to the introduction of our manuscript in line 68: “The Considère criterion is only valid for fast $\dot{\epsilon}$ compared to the characteristic relaxation time τ of the material $\dot{\epsilon}\tau \gg 1$. For pom-poms in this manuscript the inverse of the crossover frequency is used as the characteristic relaxation time, see discussion below. Corrections for strain rates where time and strain are relevant were evaluated by *Fielding et al.*”

At the bottom of page 5, the “Considere factor”, which seems to be the ratio of extensional viscosity at the stress maximum to linear viscosity at the same time point, is equal to the maximum “strain hardening factor” which is the ratio of extensional viscosity to steady state Doi-Edwards viscosity at the same strain rate, since it is defined on the bottom of page 5 as $\eta_E(\dot{\epsilon})/\eta_{DE}(\dot{\epsilon})$. This suggests that the maximum is with respect to time or strain during start-up of extension. But, in Fig. 5b, the maximum strain hardening factor is a function of strain rate presumably after finding the maximum with respect to time. Thus, the authors need to develop a clearer notation, since SHF_{max} is the maximum with respect to $\dot{\epsilon}$ of the ratio $\eta_E^{max}(\dot{\epsilon})/\eta_{DE}^{max}(\dot{\epsilon})$ where I have here introduced “max” to mean at maximum stress, which would be the peak in the viscosity curve in experiments or steady-state in the theory, right? So, there seem to be two maximizations (with respect to strain and with respect to strain rate) to obtain SHF_{max} . Also confusing is that the y axis in Fig. 5b gives these viscosities as functions of time t and strain rate, but they should be at the point of maximum stress and so do not depend on time. Is this correct? Notation is unclear.

Response: Thank you for your comment. The notification has been clarified.

In this work, the strain hardening factor is defined as $SHF = \eta_E(\dot{\epsilon})/\eta_{DE}(\dot{\epsilon})$. The Considère factor is defined against the LVE $f_c = \frac{\eta_E(\dot{\epsilon})}{\eta_{LVE}^+(1/t)} \approx \left[\frac{q^2}{\ln(\sqrt{3}q)} \right]$. In Figure A11, the LVE and the plateau viscosity of the DE model is shown as a function of angular frequency and strain rate, respectively. Judging from this, the LVE and the plateau viscosity of the DE model are in first approximation equal $\eta_{LVE}^+(1/t) = \eta_{DE}(\dot{\epsilon})$. We added this to the manuscript from line 422 on.

Figure A11: LVE and the plateau viscosity of the DE model at each investigated extensional strain rate (see Supplementary Figure 3) is shown as a function of angular frequency and strain rate, respectively.

A confusion also results from the comparison of the maximum in SHF against f_c , the Considère factor; where the former is taken as a maximum with respect to strain rate and the latter is not. Evidently this is because f_c is roughly independent of strain rate over Regime 3, and so taking a maximum is unnecessary. But, when these concepts are first introduced, it is very hard to follow what is meant. There is also the question of the experimental definition of these quantities and how they are assessed theoretically since the pom-pom theory gives extensional viscosities as a function of both extension rate and time.

Response: Thank you again for pointing out unclear notation. We clarified the manuscript in this regard. The theoretical Considère factor f_c is independent of the strain rate because it relates the extensional viscosity with the LVE at the same time/strain rate only through the arm number q . The definition of for example the strain hardening factor for the calculated values from the pom-pom model are obtained when exchanging the maximum values (experiment) to steady state values (modelling).

The following statement on page 7 is confusing: “Due to the unentangled arms, the measured plateau modulus of the sample 280k-2x30-7k is the diluted modulus of the backbone and was adjusted to $G_0 = 8 \times 10^4$ Pa to better fit the experimental LVE.” Why is this one polymer singled out for this remark? Is it the diluted modulus that was adjusted to 8×10^4 Pa? This is what the sentence seems to say, but G_0 is supposed to be the undiluted modulus. Perhaps the sentence should say: “Due to the unentangled arms, the measured plateau modulus of the sample 280k-2x30-7k is the diluted modulus of the backbone and to match the experimental LVE the undiluted modulus was adjusted to $G_0 = 8 \times 10^4$ Pa.” Also, at this point in the paper the “diluted modulus” is not defined and so its definition should be given here to help the reader understand. If this one sample is the only one for which G_0 was adjusted, is there some explanation for this exception?

Response: We thank the reviewer for raising this issue. We revised the introduction to give clearer definitions of the quantities and the calculation of the extensional data with the pom-pom model based on other raised issues. To conclude, the pom-pom model, does not give a good description of the LVE and therefore the extensional data is also not well represented. The strain hardening of the pom-poms is roughly correct. Please see Reviewer 2 comment 5) for more detailed information.

In the statement at the bottom of page 10 that “and strain hardening substantially decreases”, does “strain hardening” mean the ratio of nonlinear viscosity to linear viscosity at the same point in time? The viscosity at low strain rates looks as steep or steeper at low strains as it is at high rates, and it is only the ratio of viscosity to linear viscosity that decreases. Is this what the term “strain hardening” means? The term “strain hardening” does not seem to be defined in the text, although “strain hardening factor” is defined. Should the reader understand these two terms to be the same thing? If not, then “strain hardening” needs defining.

Response: Thank you for your remark. The term strain hardening is used when the time dependent stress growth coefficient increases over the time dependent LVE. We added to the

manuscript the definition of the term in line 39: “Strain hardening in elongational flow is the increase of the tensile stress growth coefficient over the linear viscoelastic envelope.”

Presumably the discussion on page 11 extensional viscosity curves “matching” regardless of arm molecular weight presumably means that the curves can be along the time and viscosity axes shifted into overlap, right? This should be stated more precisely than just saying that they “match.”

Response: Thank you for this further improving suggestion. The extensional viscosity values are the same for the same backbone length, arm number and strain rate irrespective of the arm molecular weight. The manuscript has been adjusted.

“If the so-called pom-pom-like rheological response as assumed in the pom-pom model is dominant within the polymer melt, the extensional viscosity is the same at a given $M_{w,b}$ and $\dot{\epsilon}$ irrespective of $M_{w,a}$. This has been predicted by multichain slip-link simulations^{43,44} and was shown for pom-poms with a backbone molecular weight of $M_{w,b} = 100 \text{ kg mol}^{-1}$ and varying molecular weight of the arms and $M_{w,a} \leq 40 \text{ kg mol}^{-1}$.³⁰ For the samples reported in this work, extensional viscosities of pom-poms at a given backbone molecular weight, arm number and a given same strain rate are the same irrespectively of the arm molecular weight ($M_{w,a} = 24$ and 40 kg mol^{-1}), as shown in Supplementary Figure 2, supporting the slip-link predictions.”

In the caption to Figure 4, “inlet” should be “inset.” The inset has no axis labels, and it is hard to know what is plotted. We are told it is the “deviation between the calculated tensile stress growth coefficient and the measured LVE,” but does this mean it is the difference $\eta_{E^+}(t) - \eta_{E^+}$, LVE (t), between the nonlinear and the linear stress growth coefficients at the time t? This would mean that the curve would tend towards zero at low strain rates. Properly labelling the x and y axes would help the reader understand what is plotted.

Response: Thank you, both mistakes have been corrected.

On page 12, the statement “overall shape of the tensile stress growth coefficient is similar to the measured tensile stress growth coefficient”. Should be “overall shape of the predicted tensile stress growth coefficient is similar to the measured tensile stress growth coefficient.

Response: Thank you, corrected.

On page 12, I don't understand what is meant by “the highest measured extensional viscosity is predicted well only from the topology.” This seems to mean that only q, the number of arms is required to obtain the highest measured viscosity? While the Considered ratio f_c depends only on q, f_c is the extensional viscosity over its linear viscosity and the latter depends on other factors besides q, doesn't it? The statement that “the predictions overestimate the LVE between $\sim 1 - 100 \text{ s}$, by up to 50 %.” can't be assessed because the time axis on the insert is not labelled. I don't understand what is meant by “a brief decline in the slope of the measured LVE”. Where do I see this reduced slope? Is it reduced with respect to the predicted slope? It is hard to understand what the authors are talking about. The top of page 13 refers to sample 280k-2x30-7k, saying that the stress growth coefficient for this sample is predicted more

accurately than for “280k-2x30-7k and 100k-2x11-9k”. But the first of these two samples, 280k-2x30-7k, is the same as the sample which it is said to be more accurate. So, there is some labeling mix-up here. The statement “The prediction of the strain rate with the highest measured extensional viscosity is the closest to the measured tensile stress growth coefficient,” seems only valid for Fig. 4a, and seems to be referring to the curves for strain rate 0.0056; is this correct? This could be made clearer.

Response: Thank you for your careful revision. We revised the part of the manuscript according to your comments and from the other reviewers, during which the above-mentioned parts have been removed.

The statement “Equation (13) seems to systematically overestimate the Weissenberg number of the strain hardening onset, suggesting a difference in the relaxation mechanism for combs and pom-poms.” To be clear, do you mean that for combs, the strain hardening start at Wi much above unity, while for these pom-poms, hardening starts at Wi near unity?

Response: Yes, we speculate so. Systematic investigation of the molecular parameters of combs is currently conducted.

I don't understand the claim “Stress release of a (partially) stretched backbone can in this picture only be achieved by the reptation of the whole pom-pom molecule.” The stretch relaxation time of the pom-pom backbone only requires Rouse motion of the backbone within the tube, which occurs at a rate set by the mobility of the branch points and is not influenced by number of entanglements, while complete orientational relaxation of the backbone requires reptation of the backbone out of the tube. Reptation which should be a factor of Z_{diluted} slower than the Rouse relaxation, where “ Z_{diluted} ” is the diluted number of tube segments in the backbone. The value of Z_{diluted} for the pom-pom 280k-2x22-22k would seem to be around unity, since it is proportional to the square of the volume fraction of backbone. So, possibly the reason strain hardening occurs at $Wi = 1$, is because the backbone is not self-entangled for this pom-pom. Could similar reasoning be applied to the other pom-poms to see if the onset of strain hardening for them can be explained? In some of them, Z_{diluted} is presumably significantly greater than unity, and they should show strain hardening $Wi > 1$. If this reasoning is correct, then it seems that the analogous combs with same arm and backbone molecular weights, and same number of branches, may require higher Wi to strain harden significantly than for pom-poms because only the most central backbone segment of the comb feels the full stretch possible, since other segments have fewer arms on one side of them and thus do not stretch as much.

Response: Thank you for your comment. We agree that a hierarchical stress distribution on the backbone of combs is likely. For our pom-poms, we think that the backbone stretch relaxation time is about the inversion of the crossover frequency into the terminal regime, because all samples show the onset of strain hardening around $Wi = 1$. We will conduct comparison between combs and pom-poms with the same molecular parameter in the future to see if differences in the relaxation times can be discovered due to the different positioning of the chains.

As stated in the manuscript in line 339, $\tau_l/\tau_R \propto s$ for linear chains. With the longest relaxation time for pom-poms being the orientation relaxation time and taking the stretch relaxation time τ_s of the pom-poms as the Rouse time τ_R for linear chains, we can estimate $s_{b,diluted}$ for our pom-poms with the aforementioned relation. The orientation relaxation time can be estimated from the onset of shear thinning. For 280k-2x22-22k, we can estimate $s_{b,diluted} \sim \frac{1 Wi}{0.1 Wi} \sim 10$ resulting in a reduction of around 7 entanglements.

These sentences are confusing: “the extensional viscosity is only depending on the backbone length. Together with the Considère limit, these findings suggest that strain hardening of pom-pom systems depends only on the number of arms attached to the branchpoints and not on the length of the arms or the backbone.” The first sentence says that the extensional viscosity depends only on backbone length and the second that strain hardening does not depend on backbone length. Again, the term “strain hardening” is not precisely defined, so it is hard to know precisely what is meant. But somehow the definition of strain hardening allows it to not depend on backbone length although the viscosity does depend on it? This is confusing.

Response: Thank you for pointing this out. We clarified in the manuscript from line 367 on: “As shown in Supplementary Figure 2 and discussed earlier, the extensional viscosity is only depending on the backbone length and arm number, independent of $M_{w,a}$. Together with the Considère limit, these findings suggest that strain hardening (factor) ($SHF = \eta_E(\dot{\epsilon})/\eta_{DE}(\dot{\epsilon})$) of pom-pom systems depends only on the number of arms attached to the branchpoints and not on the length of the arms or the backbone. This SHF is given by the Considère limit as a function of the arm number.”

The scaling law of extensional viscosity with -0.5 exponent is not explained, and yet is crucial to the prediction of the Considère criterion. If the pom-pom model is used, the linear rheology does not show a -0.5 power law, and this must arise from dynamics not in the pom-pom model. But the derivation of the Considère factor after Eq. 11 requires using the pom-pom linear rheology formula Eq. 6, which does not follow a power law of -0.5, and substituting its low-time expansion into Eq. 5. But the Considère’ criterion is related to the nonlinear rheology, not the linear rheology and so this substitution seems to be cancelling out an error in the prediction of the nonlinear behavior by a proportional error in the linear rheology. Why should we consider the Considère criterion expressed as a ratio to a linear response the most valid expression? The authors should discuss this and at least note that the expression for the linear behavior is not accurate and this has implications for the validity of Eq. 11 for their data.

Response: Thank you for raising this important issue. There are two things to be considered here:

The description of the LVE through the pom-pom model is indeed needed for the derivation of Eq. 11. In Eq. 11, the time dependent LVE viscosity η_{LVE}^+ is then substituted by the experimental LVE as shown in Figure 5a). The slope of -0.5 then results from the experimental LVE data, because the Considère criterion only relates η_E to η_{LVE}^+ at a given time, independent of the slope of η_E or η_{LVE}^+ as a function of time.

In an ideal model, e.g., the pom-pom model, a slope of -1 is expected.² In the Doi-Edwards model, which can predict the rheology of linear chains, the slope is reduced to -0.8.³ The slope in the experimental LVE and extensional viscosity is roughly -0.5. We think that is a result of

the superimposed relaxation behavior of the arms and the backbone leading to the decreased slope of -0.5 as well as a combination of chains reaching their stretch through finite extensibility and friction reduction, as pointed out by reviewer 2.

Are the orange data points given in Fig. 6 experimental data points? If so, the sample should be named in the caption. If not, please note in the caption that they are schematic.

Response: Thank you, corrected. We added to the figure caption: “Idealized LVE and extensional viscosities in the light blue line and the orange squares, respectively.”

The summary discusses Considère factor and maximum strain hardening factor, but these should be defined clearly also in the summary so that the reader who skims the article or forgets the definitions is refreshed.

Response: Thank you, corrected.

Minor points:

Since I am not an expert on organic synthetic chemistry, I confine my comments to the rheological portion of the paper.

Response: The synthesis route towards the pom-poms samples is robust and yields pom-pom material in 10 - 30 g range with high purity and narrow molecular weight distribution ($PDI < 1.2$) within several days. It has been reviewed multiple times and was published in two peer-reviewed publications beforehand, also in more polymer chemistry orientated journals.^{4,5} A further optimization is currently investigated and should be published shortly after.

In Eq. 1, the stress tensor σ should be bold since it is a tensor.

Response: Thank you, corrected.

References

1. van Ruymbeke, E., Kapnistos, M., Vlassopoulos, D., Huang, T. & Knauss, D. M. Linear Melt Rheology of Pom-Pom Polystyrenes with Unentangled Branches. *Macromolecules* **40**, 1713–1719; 10.1021/ma062487n (2007).
2. McLeish, T. C. B. & Larson, R. G. Molecular constitutive equations for a class of branched polymers: The pom-pom polymer. *Journal of Rheology* **42**, 81–110; 10.1122/1.550933 (1998).
3. Doi, M. & Edwards, S. F. *The theory of polymer dynamics* (Clarendon Press, Oxford, 2013).
4. Röpert, M.-C., Schußmann, M. G., Esfahani, M. K., Wilhelm, M. & Hirschberg, V. Effect of Side Chain Length in Polystyrene POM–POMs on Melt Rheology and Solid Mechanical Fatigue. *Macromolecules*; 10.1021/acs.macromol.2c00199 (2022).

5. Röpert, M.-C., Hirschberg, V., Schußmann, M. G. & Wilhelm, M. Impact of Topological Parameters on Melt Rheological Properties and Foamability of PS POM-POMs. *Macromolecules*; 10.1021/acs.macromol.2c02051 (2023).

Reviewers' Comments:

Reviewer #1:

Remarks to the Author:

[Note from the Editor: Reviewer #1 was also asked to look over the response given to reviewer #2]

The authors have addressed all my questions. I have a few further comments as follows.

1. Regarding my previous Comment 1 (about the Tg of the samples):

It is interesting that the Tg is not significantly influenced by the short arms. But as shown in Fig.A1 there is still a difference up to 2.8°C – is it due to a few remaining short (free) chains/residual solvents in the samples?

2. Regarding my previous Comment 2 (about the rubber plateau of 100k-2x12-24k):

I am still not convinced that the backbones of the sample 100k-2x12-24k is not entangled. The authors replied that “The rubber plateau is the same for all samples with the same number and molecular weight of the arms and irrespective of the backbone molecular weight”, but how about the rubber plateau for samples with the same backbone 100k and different number/Mw of arms? I think the 100k linear PS (i.e., without any arms) also has a similar rubber plateau.

3. Regarding my previous Comment 3 (about the real Hencky strain in EVF):

I just have a comment here about the ratio of width to length: It seems to me that the authors took the whole length of the specimen for the calculation, but in the reference the length is only the part between the axis of the two drums (in SER it is 12.7mm; in EVF probably the same). Therefore, the width to length ratio of the samples in this manuscript is around 0.4 – it might have some influence but probably not change the conclusion.

4. In the author’s reply to Reviewer #2, different relaxation times are discussed. But I am still confused about these relaxation times after reading the revised manuscript:

1) As mentioned by the authors, in the pom-pom model there are two relaxation times related to the backbone- the orientation relaxation time τ_b , and the stretch relaxation time τ_s . In the original pom-pom model, τ_b and τ_s are related to the arm relaxation time τ_a . In the manuscript, the authors used the Rouse time of the entangled segments τ_e for the related calculations, but the relationship between τ_e and τ_s (or τ_b) is not clearly described.

2) How the τ_e is determined? Is it taken from the inverse frequency of the crossover point (the one at the higher frequency)? On line 193, τ_e is 0.036 s; on line 263, it is 0.0036 s – one of them must be a typo.

3) The authors used the longest relaxation time τ_l to define the W_i in Fig.5. And then on line 315-317 the authors wrote that “Between $\sim 0.3 < W_i < 1$, shear and extensional thinning can be observed, ...caused by orientation of backbone chains due to the flow faster than the orientation relaxation time τ_b .” What is the relationship between τ_l and τ_b ? Are they equivalent to each other?

Reviewer #3:

Remarks to the Author:

[Note from the Editor: Reviewer #3 was also asked to look over the response given to reviewer #2]

Second review of Predicting Maximum Strain Hardening Factor in Elongational Flow of Branched Pom-Pom Polymers from Polymer Architecture

The paper has been improved considerably, but still has confusing aspects.

The paper now has a section in the Introduction that defines the backbone orientation and stretch relaxation times τ_b and τ_s , respectively. The retraction of the arms is said to take place at the "time t_q ". Does "time" here literally mean "time" rather than a relaxation time? If time is to be taken literally, then is this the time after start of extensional flow? But the sentence that follows is confusing: "Therefore, the pom-pom model is limited to times faster than the arm relaxation time τ_a as well as the Rouse relaxation time τ_R , because at Hencky strain rates $\dot{\epsilon} > 1/\tau_R$ all polymer chains show strain hardening due to monomer friction. For slow times, it is limited to $\dot{\epsilon} > 1/\tau_b$ ". By "limited to times faster than," does this mean "limited to strain rates higher than" the inverse of τ_a ? Or does it really mean times longer than τ_a ? Then a new time, the "Rouse relaxation time" is introduced, but not defined. Is this the Rouse relaxation time of the backbone? If so, isn't this the same as τ_s ? By "all chains show strain hardening," is this because of stretch of the backbone? This would occur for strain rates greater than $1/\tau_s$. What does "slow times" mean; is it literally long times, or is it low strain rates? If it means low strain rates, then are these strain rates limited by the criterion that $\dot{\epsilon} > 1/\tau_b$? If so, we seem to be given two requirements, namely that the pom-pom model is limited to strain rates faster than $1/\tau_a$, and also faster than $1/\tau_b$. But this makes little sense, and the added material has just made everything more confusing rather than less so.

Just after Eq. 3, the new material added implies that extensional stress is given by the symbol F . But stress is given by the symbol σ elsewhere, such as in Eq. 1.

Eq. 4 gives a formula for ϵ^+_{+E} as a function of t and ϵ , taken to the limit in which ϵ_f goes to ϵ_q . Is this a limit of large $\dot{\epsilon}$ or large time, or both? The right side of the equation depends on neither time t nor strain rate $\dot{\epsilon}$. So is the limit taken on the left side a double limit so that result on the right is independent of time and strain rate? Where does Equation 4 come from? Is this from the pom-pom model? If so, how did the factor of $15/4$ go away? The "limiting case" leading to Eq. 5 has already been taken in Eq. 4, since Eq. 5 is the same as Eq. 4, except that η^+_{+E} has been replaced by η_{+E} .

The time t_q is defined as the time to stretch the backbone to the extension needed to pull in the arms, and is thus the time to reach maximum stretch of the backbone. Is this time assumed to be under conditions of affine stretching; i.e., no relaxation during the stretch?

The new material added in line 182 is confusing: "Generally, for $q > 22$, at high rates the sample fails at $\epsilon < 4$, at medium strain and low rates the sample does not fail beforehand." The statement "does not fail beforehand" should presumably be replaced by "does not fail until strains higher than 4." Is this what is intended. Then the statement is "However, for all samples the stress maximum is reached for $\epsilon < 4$." So, the stress maximum is reached for $\epsilon < 4$, but the sample only fails when $\epsilon > 4$; is this what is intended? If so, how do you know that the sample is not failing if a stress maximum has been reached? Isn't the stress maximum a sign that the sample is "failing"? Then, "For $q \leq 13$, samples fail at $\epsilon < 4$." Does this last statement apply for any strain rate? Is this how the $q > 22$ condition differs from when $q < 13$?

In the response to reviewer 2, "Figure A" is used repeatedly, but seems to mean "Figure A5." Supplementary Figure 3 is also referred to, but I don't see its connection to Figure A5 shown in the response. There are also "Figure A6" and "Figure A7" in the response to reviewer 2, but they seem to be referred to as "Figure A".

On page 10, the statement is puzzling: "If the so-called pom-pom-like rheological response as assumed in the pom-pom model is dominant within the polymer melt, the extensional viscosity is at a given $M_{w,b}$, q , and ϵ the same, irrespective of $M_{w,a}$. This is puzzling, because Eq. 3 shows a dependence of extensional viscosity on s_b and s_a , implying a dependence on $M_{w,a}$."

On page 11, the comment is confusing: "For the samples reported in this work, extensional viscosities

of pom-poms at a given backbone molecular weight, arm number and a given strain rate are the same irrespectively of the arm molecular weight ($M_{w,a} = 24$ and 40 kg mol^{-1}), as shown in Supplementary Figure 2, supporting the slip-link predictions." The arm molecular weight affects the volume fraction of backbone, which should affect the viscosity, should it not? I see in supplementary Figure 3, that for a backbone of 100k, an increase in arm molecular weight from 9k to 24k to 40k leads to a clear increase in viscosity, especially the plateau values, in the data and also to some extent in the predictions. Do the authors want to correct their statement about the lack of dependence of extensional viscosity on arm molecular weight? A similar claim is made on page 15, which does not seem to be generally valid: "As shown in Supplementary Figure 2 and discussed earlier, the extensional viscosity is only depending on the backbone length and the arm number, independent of $M_{w,a}$."

The caption to Fig. 4 is does not well explain what is in the figure. Evidently, the solid, colored lines in part a use the pom-pom formulas for the parameters, while the solid, colored lines in b use an larger modulus than in is used in part a, correct? This should be made clearer in the legend, perhaps by a statement in b that the solid colored lines use adjusted G_0 . The legend to part b indicates that the short dashed lines are for "LVE adjusted", but this is actually the predicted LVE with parameters adjusted, right? If so, this should be made clear in the legend or caption.

In the caption to Figure 5, "normalized to" should be "normalized by".

On page 14, the statement "considering increased monomeric friction of the branches" should probably be "considering the additional monomeric friction contributed by the branches"

I don't understand the reply to the review that says "As you estimated, $\tau_{R,eq.13}$ should be slightly smaller than $\tau_{R, pom-pom}$." $\tau_{R,eq.13}$ is presumably " $\tau_{R,br}$ " given by Eq. 13. What is $\tau_{R, pom-pom}$ " and how does it differ from τ_{R} ?

The statement on page 14: "a Rouse time for branched systems $\tau_{R,br} = 44.6 \text{ s}$ can be calculated, which corresponds with $\tau_{l} = 1672 \text{ s}$ to W_i ($\tau_{R,br} = 37.5$ " is confusing. To compute W_i , an extension rate is needed; what extension rate is to be used? Is this extension rate the inverse of $\tau_{R,br}$? If so, this should be made clearer. Does $W_i = 37.5$ mean that there are 37.5 backbone entanglements in the sample?

On page 15, the following is confusing: "the pom-pom model is only valid for $\dot{\epsilon} > 1/\tau_{R}$, which is estimated to be in the lower bc range of regime III. Therefore, the model is in principle not valid in the higher bc range of regime III." If the condition for validity of the pom-pom model is $\dot{\epsilon} > 1/\tau_{R}$, how does this lead to the conclusion that the validity of the pom-pom model is limited to lower strain rates? What limits its application to higher strain rates? Or should the limit of validity of the pom-pom model be $\dot{\epsilon} < 1/\tau_{R}$ rather than the reverse? Please check the statements carefully.

On page 19, "scale quadratic" should be "scale quadratically."

In a response to a question about "strain hardening," the review responds that "the term strain hardening is used when the time dependent stress growth coefficient increases over the time dependent LVE." The "strain hardening factor" however is the ratio of stress to Doi-Edwards stress at steady state at the same strain rate, as defined on page 4 of the manuscript. It is odd that "strain hardening" which seems to be the ratio of stress over the stress in the linear limit at the same time, and is a time-dependent quantity, has a different definition than "strain hardening factor." Perhaps the authors should use the term "time-dependent strain hardening, or make it clearer the difference between these.

In Figure 7, the caption refers to the backbone concentration is given as " ϕ_{bb} " while on the plot the x axis in part b gives it as " ϕ_b ". Figure 7a is supposed to be a plot against number of backbone

entanglements, but on the figure, the top two curves are labelled as "100k-2x12-24k," for regimes I/II, suggesting that all points on these curves are for a backbone of 100k. This does not seem possible, since the x axis is the length of backbone. Should the "100k" in the legend be replaced by "M_{w,b}" to indicate that the backbone molecular weight is variable along the curve?

In the summary, the "Rouse time" should be defined there, even if it is defined earlier (I'm not sure that it is).

In the response to my point (Reviewer 3) that "The scaling law of extensional viscosity with -0.5 exponent is not explained," the authors give an explanation of their thinking. This explanation for the -0.5 exponent, rather than -1, should be mentioned in the text somewhere.

REVIEWER COMMENTS

Reviewer #1 (Remarks to the Author):

[Note from the Editor: Reviewer #1 was also asked to look over the response given to reviewer #2]

The authors have addressed all my questions. I have a few further comments as follows.

1. Regarding my previous Comment 1 (about the T_g of the samples):

It is interesting that the T_g is not significantly influenced by the short arms. But as shown in Fig.A1 there is still a difference up to 2.8°C – is it due to a few remaining short (free) chains/residual solvents in the samples?

Yes, it is indeed interesting. We think the difference is attributed to measurement inaccuracy. An interlaboratory study revealed for PMMA ($T_g = 103^\circ\text{C}$, same magnitude as PS) a repeatability limit of 2°C within different runs on the same instrument and sample.¹ In our case, we have different samples with different M_w and M_w of the arms, resulting in a lower repeatability and including the difference in our samples a T_g difference of up to 2.8°C seems reasonable.

2. Regarding my previous Comment 2 (about the rubber plateau of 100k-2x12-24k):

I am still not convinced that the backbones of the sample 100k-2x12-24k is not entangled. The authors replied that “The rubber plateau is the same for all samples with the same number and molecular weight of the arms and irrespective of the backbone molecular weight”, but how about the rubber plateau for samples with the same backbone 100k and different number/Mw of arms? I think the 100k linear PS (i.e., without any arms) also has a similar rubber plateau.

The rubber plateaus of samples with 100 kg mol^{-1} backbone and increasing arm molecular weight as well as linear 100k PS are shown below in Figure A 1. Indeed, the rubber plateaus are fairly similar to each other. As expected, an increase in the arm relaxation time with increasing arm molecular weight can be observed.

Figure A 1: Arm rubber plateau of pom-poms with 100 kg mol^{-1} backbone as well as 100k linear PS.

We must revise our previous response to: “The rubber plateau of the arms is the same for all samples with the same number and molecular weight of the arms and irrespective of the backbone molecular weight.” to include “of the arms”. Further we clarify, that at high frequencies the backbone is very much entangled with the arms resulting in the observed rubber plateau. Our statement was aimed at lower frequencies where the low frequency phase angle minimum can be found. Here the backbone is

not self-entangled, resulting in no secondary rubber plateau due to dynamic tube dilution. We revised the section in the manuscript (line 201f.).

3. Regarding my previous Comment 3 (about the real Hencky strain in EVF):

I just have a comment here about the ratio of width to length: It seems to me that the authors took the whole length of the specimen for the calculation, but in the reference the length is only the part between the axis of the two drums (in SER it is 12.7mm; in EVF probably the same). Therefore, the width to length ratio of the samples in this manuscript is around 0.4 – it might have some influence but probably not change the conclusion.

Thank you for the correction. We indeed took the sample length for the calculation of the length ratio but agree with you that this does not change the conclusion.

4. In the author's reply to Reviewer #2, different relaxation times are discussed. But I am still confused about these relaxation times after reading the revised manuscript:

1) As mentioned by the authors, in the pom-pom model there are two relaxation times related to the backbone- the orientation relaxation time τ_b , and the stretch relaxation time τ_s . In the original pom-pom model, τ_b and τ_s are related to the arm relaxation time τ_a . In the manuscript, the authors used the Rouse time of the entangled segments τ_e for the related calculations, but the relationship between τ_e and τ_s (or τ_b) is not clearly described.

To this point we are unaware of any relationship between τ_e , τ_s and τ_b for the pom-pom topology. We use the Rouse time of the whole pom-pom molecule to estimate the high strain rate limit because we know that at higher strain rates monomer friction between all chains contributes to strain hardening and we can approximately calculate the Rouse time by the relaxation from Eq.13 (Lentzakis et al. 2013).

2) How the τ_e is determined? Is it taken from the inverse frequency of the crossover point (the one at the higher frequency)? On line 193, τ_e is 0.036 s; on line 263, it is 0.0036 s – one of them must be a typo.

Thank you, we corrected τ_e in line 263 to 0.036 s. We adapted τ_e from *Abbasi et. al., Macromolecules* 2017, in which the authors use the intersection between the slope of 0.5 in G' and the plateau modulus of G' .

3) The authors used the longest relaxation time τ_l to define the Wi in Fig.5. And then on line 315-317 the authors wrote that "Between $\sim 0.3 < Wi < 1$, shear and extensional thinning can be observed, ...caused by orientation of backbone chains due to the flow faster than the orientation relaxation time τ_b ." What is the relationship between τ_l and τ_b ? Are they equivalent to each other?

Thank you for your comment. We attributed τ_l to the crossover frequency into the terminal regime due to easy and precise determination, see also paragraph starting from line 290. From the observed onset of strain hardening around $Wi = 1$, we can approximate $\tau_l \approx \tau_s$.

McLeish and Larson relate the stretch relaxation time τ_s to the orientation relaxation time τ_b through $\tau_b = 4/\pi^2 s_b \phi_b \tau_s \sim 4/\pi^2 s_b \phi_b \tau_l$. Therefore, τ_s and τ_b are not identical, and τ_b is larger than τ_s which we find from the experimental findings to equal in first approximation to τ_l , i.e. $\tau_l \approx \tau_s$.

Reviewer #3 (Remarks to the Author):

[Note from the Editor: Reviewer #3 was also asked to look over the response given to reviewer #2]

Second review of Predicting Maximum Strain Hardening Factor in Elongational Flow of Branched

Pom-Pom Polymers from Polymer Architecture

The paper has been improved considerably, but still has confusing aspects.

The paper now has a section in the Introduction that defines the backbone orientation and stretch relaxation times τ_b and τ_s , respectively. The retraction of the arms is said to take place at the “time t_q ”. Does “time” here literally mean “time” rather than a relaxation time? If time is to be taken literally, then is this the time after start of extensional flow?

Yes, t_q is a literal time and taken after the start of the extensional flow. Note that t_q is a specific time at a set strain rate and $t_q \sim \dot{\epsilon}$. We adapted Figure 2a from McLeish and Larson to illustrate t_q below.

Figure A 2: Figure 2a adapted from Larson, McLeish 1999 to illustrate t_q in the tensile stress growth coefficient. The strain rates are 0.04, 0.02, and 0.01 in respect to the arm relaxation time $\tau_a(0)$.

But the sentence that follows is confusing: “Therefore, the pom-pom model is limited to times faster than the arm relaxation time τ_a as well as the Rouse relaxation time τ_R , because at Hencky strain rates $\epsilon \dot{> 1/\tau_R}$ all polymer chains show strain hardening due to monomer friction. For slow times, it is limited to $\epsilon \dot{> 1/\tau_b}$ ”. By “limited to times faster than,” does this mean “limited to strain rates higher than” the inverse of τ_a ? Or does it really mean times longer than τ_a ?

Thank you for the suggestion. Yes, the pom-pom model is limited to strain rates smaller than $\frac{1}{\tau_a}$. We adjusted the manuscript accordingly for more clarity, please see line 61f.

Then a new time, the “Rouse relaxation time” is introduced, but not defined. Is this the Rouse relaxation time of the backbone? If so, isn’t this the same as τ_s ? By “all chains show strain hardening,” is this because of stretch of the backbone? This would occur for strain rates greater than $1/\tau_s$.

The Rouse time is the stretch relaxation time of linear polymers. In other words, at higher strain rates than the inverse of the Rouse time, even linear polymers show strain hardening. We included a reference for clarity. In the context of branched chains, the Rouse time corresponds to the whole molecule, including arms and backbone. We unified all references to Rouse time to increase clarity.

What does “slow times” mean; is it literally long times, or is it low strain rates? If it means low strain rates, then are these strain rates limited by the criterion that $\epsilon \dot{> 1/\tau_b}$? If so, we seem to be given two requirements, namely that the pom-pom model is limited to strain rates faster than $1/\tau_a$, and also faster than $1/\tau_b$. But this makes little sense, and the added material has just made everything more confusing rather than less so.

Thank you for your comment, we exchanged all times to strain rates to improve readability and precision. To summarize the limits, we added: “To summarize, the pom-pom model is valid for strain rates $\frac{1}{\tau_R} > \dot{\epsilon} > \frac{1}{\tau_b}$ or in case of $\frac{1}{\tau_a} > \frac{1}{\tau_R}$ it is $\frac{1}{\tau_a} > \dot{\epsilon} > \frac{1}{\tau_b}$.”

Just after Eq. 3, the new material added implies that extensional stress is given by the symbol F. But stress is given by the symbol sigma elsewhere, such as in Eq. 1.

Thank you, corrected.

Eq. 4 gives a formula for ϵ^+_{E} as a function of t and epsilon, taken to the limit in which ϵ_{f} goes to ϵ_{q} . Is this a limit of large epsilon dot or large time, or both? The right side of the equation depends on neither time t nor strain rate epsilon dot. So is the limit taken on the left side a double limit so that result on the right is independent of time and strain rate? Where does Equation 4 come from? Is this from the pom-pom model? If so, how did the factor of 15/4 go away? The “limiting case” leading to Eq. 5 has already been taken in Eq. 4, since Eq. 5 is the same as Eq. 4, except that ϵ^+_{E} has been replaced by ϵ_{E} .

In the experimental part of this manuscript and the application of the Considère limit onto the pom-pom model, uniaxial extension at a constant Hencky strain rate is considered, thus it is not a limit of the strain rate, but the strain rate rather refers to the measurement condition used. For the time: the Hencky strain ϵ is given by the applied strain rate and the duration $\epsilon = \dot{\epsilon}t$. Therefore, when taking the limit $\epsilon_{\text{f}} \rightarrow \epsilon_{\text{q}}$, the limit yields a specific time. The factor 15/4 as well as other prefactors such as G_0 are aggregated into η_0 . Eq 5 shows the extensional viscosity at $\epsilon_{\text{f}} = \epsilon_{\text{q}}$. We adjusted the manuscript in line 81f. to describe the transformation between Eq. 4 and Eq. 5 more clearly.

The time t_{q} is defined as the time to stretch the backbone to the extension needed to pull in the arms and is thus the time to reach maximum stretch of the backbone. Is this time assumed to be under conditions of affine stretching; i.e., no relaxation during the stretch?

Yes, affine stretching is assumed. The pom-pom model specifies $\tau_a < t < \tau_b$ for its validity, therefore no relaxation of the backbone takes place and all arms are relaxed during the time.

The new material added in line 182 is confusing: “Generally, for $q > 22$, at high rates the sample fails at $\epsilon < 4$, at medium strain and low rates the sample does not fail beforehand.” The statement “does not fail beforehand” should presumably be replaced by “does not fail until strains higher than 4.” Is this what is intended. Then the statement is “However, for all samples the stress maximum is reached for $\epsilon < 4$.” So, the stress maximum is reached for $\epsilon < 4$, but the sample only fails when $\epsilon > 4$; is this what is intended? If so, how do you know that the sample is not failing if a stress maximum has been reached? Isn't the stress maximum a sign that the sample is “failing”? Then, “For $q \leq 13$, samples fail at $\epsilon < 4$.” Does this last statement apply for any strain rate? Is this how the $q > 22$ condition differs from when $q < 13$?

Thank you for your thorough review. The intention was to say that the sample does not fail until a strain of 4. The statement has been corrected. Indeed, it was intended as you understood it. For $q > 22$, the samples typically do not fail until $\epsilon > 4$ for medium and low strain rates. This can be easily confirmed by measurement data of the extensional viscosity, although not valid due to sample overlap, and visual inspection of the specimen after extension to $\epsilon = 4$. Typically, a plateau viscosity is reached around Hencky strain of 4.

The mentioned statements have been corrected.

In the response to reviewer 2, “Figure A” is used repeatedly, but seems to mean “Figure A5.” Supplementary Figure 3 is also referred to, but I don't see its connection to Figure A5 shown in the response. There are also “Figure A6” and “Figure A7” in the response to reviewer 2, but they seem to be referred to as “Figure A”.

Thank you for your note and please excuse the formatting mistake from our side.

On page 10, the statement is puzzling: “If the so-called pom-pom-like rheological response as assumed in the pom-pom model is dominant within the polymer melt, the extensional viscosity is at a given M_w , q , and ϵ the same, irrespective of M_w ”, This is puzzling, because Eq. 3 shows a dependence of extensional viscosity on s_b and s_a , implying a dependence on M_w . On page 11, the comment is confusing: “For the samples reported in this work, extensional viscosities of pom-poms at a given backbone molecular weight, arm number and a given strain rate are the same irrespectively of the arm molecular weight ($M_w = 24$ and 40 kg mol⁻¹), as shown in Supplementary Figure 2, supporting the slip-link predictions.” The arm molecular weight affects the volume fraction of backbone, which should affect the viscosity, should it not? I see in supplementary Figure 3, that for a backbone of 100k, an increase in arm molecular weight from 9k to 24k to 40k leads to a clear increase in viscosity, especially the plateau values, in the data and also to some extent in the predictions. Do the authors want to correct their statement about the lack of dependence of extensional viscosity on arm molecular weight? A similar claim is made on page 15, which does not seem to be generally valid: “As shown in Supplementary Figure 2 and discussed earlier, the extensional viscosity is only depending on the backbone length and the arm number, independent of M_w .”

Thank you for your in-depth comment. When examining Eq. 3 with the backbone volume fractions of our samples, i.e., 400k-2x12-23k $\phi_b^2 = 0.42^2 = 0.1764$ and 400k-2x13-40k with $\phi_b^2 = 0.28^2 = 0.0784$, the arm molecular weights and therefore volume fractions result in a factor of 2.25 difference in η_E .

$$\eta_E = \frac{15}{4} G_0 \phi_b^2 q^2 \dot{\epsilon}^{-1}$$

This factor from the different volume fraction is too small to make a reliable statement. Therefore, to determine a reliable trend of η_E as a function of the arm molecular weight, a higher difference in arm molecular weight needs to be considered here and will be investigated in the future. Furthermore, it is difficult to compare the extensional data of different pom-pom samples: the arms need to be relaxed ($\dot{\epsilon} > 1/\tau_a$), but the backbone is needed to be unrelaxed ($\dot{\epsilon} < 1/\tau_s$). Due to this uncertainty, we removed our statement.

The caption to Fig. 4 is does not well explain what is in the figure. Evidently, the solid, colored lines in part a use the pom-pom formulas for the parameters, while the solid, colored lines in b use a larger modulus than in is used in part a, correct? This should be made clearer in the legend, perhaps by a statement in b that the solid colored lines use adjusted G_0 . The legend to part b indicates that the short dashed lines are for “LVE adjusted”, but this is actually the predicted LVE with parameters adjusted, right? If so, this should be made clear in the legend or caption.

Thank you, corrected and the caption of Figure 4 is revised.

In the caption to Figure 5, “normalized to” should be “normalized by”.

Thank you, corrected.

On page 14, the statement “considering increased monomeric friction of the branches” should probably be “considering the additional monomeric friction contributed by the branches”.

Thank you, corrected.

I don't understand the reply to the review that says “As you estimated, $\tau_{R,eq.13}$ should be slightly smaller than $\tau_{R, pom-pom}$.” $\tau_{R,eq.13}$ is presumably “ $\tau_{R,br}$ ” given by Eq. 13. What is $\tau_{R, pom-pom}$ and how does it differ from τ_{R} ?

In general, the Rouse time depends not only on the number of entanglements in a polymer, but also on the topology, as Eq.13 for a comb polymer differs from the formular for a linear polymer, given by $\tau_R = z^2 * \tau_e$ (for a linear polymer only). Consequently, to this point we are unaware of any exact

equation for the Rouse time of a pom-pom $\tau_{R,pom-pom}$. We estimate that the Rouse time of a comb, shown in $\tau_{R,eq.13}$ should be slightly smaller than the Rouse time of a pom-pom, $\tau_{R,pom-pom}$, due to the increased inertia for pom-poms from moving the equally spaced branches of a comb to the ends as also suggested by reviewer 2. Thus Eq. 13 is still the best estimation of the Rouse time of a pom-pom and should estimate it in first approximation correctly.

The statement on page 14: “a Rouse time for branched systems $\tau_{R,br} = 44.6$ s can be calculated, which corresponds with $\tau_l = 1672$ s to $Wi(\tau_{R,br}) = 37.5$ ” is confusing. To compute Wi , an extension rate is needed; what extension rate is to be used? Is this extension rate the inverse of $\tau_{R,br}$? If so, this should be made clearer. Does $Wi = 37.5$ mean that there are 37.5 backbone entanglements in the sample?

Correct, a strain rate is needed to calculate Wi . The Weissenberg number $Wi = 37.5$ is calculated based on the relation $Wi = \tau_l * \dot{\epsilon}_{R,br} = \tau_l * \frac{1}{\tau_{R,br}}$. We adjusted the equation accordingly.

On page 15, the following is confusing: “the pom-pom model is only valid for $\epsilon \dot{>} 1/\tau_{R,br}$, which is estimated to be in the lower bc range of regime III. Therefore, the model is in principle not valid in the higher bc range of regime III.” If the condition for validity of the pom-pom model is $\epsilon \dot{>} 1/\tau_{R,br}$, how does this lead to the conclusion that the validity of the pom-pom model is limited to lower strain rates? What limits its application to higher strain rates? Or should the limit of validity of the pom-pom model be $\epsilon \dot{<} 1/\tau_{R,br}$ rather than the reverse? Please check the statements carefully.

Thank you for your careful revision. $\dot{\epsilon} < 1/\tau_{R,br}$ is correct as the strain rates need to be smaller than the inverse Rouse time for strain hardening to result from the branched topology and not monomer friction.

On page 19, “scale quadratic” should be “scale quadratically.”

Thank you, corrected.

In a response to a question about “strain hardening,” the review responds that “the term strain hardening is used when the time dependent stress growth coefficient increases over the time dependent LVE.” The “strain hardening factor” however is the ratio of stress to Doi-Edwards stress at steady state at the same strain rate, as defined on page 4 of the manuscript. It is odd that “strain hardening” which seems to be the ratio of stress over the stress in the linear limit at the same time, and is a time-dependent quantity, has a different definition than “strain hardening factor.” Perhaps the authors should use the term “time-dependent strain hardening or make it clearer the difference between these.

Strain hardening is experimentally determined as the increase of the tensile stress growth under a non-linear, steady rate elongation above the LVE (measured in shear, in the linear regime and measured mostly under oscillatory conditions) in the time dependent representation. The strain hardening factor quantifies the maximum amount of strain hardening at a specific strain rate and is therefore not time dependent. The strain hardening factor is – as also commonly defined in literature²⁻⁹ – as the ratio of the tensile stress growth coefficient in the steady state value to the steady state value of the Doi-Edwards model in elongational flow (at the same strain rate).

Hence, the Strain hardening factor is a strain rate dependent quantity, as it quantifies the maximum (time dependent) strain hardening for a specific strain rate.

In Figure 7, the caption refers to the backbone concentration is given as “ ϕ_{bb} ” while on the plot the x axis in part b gives it as “ ϕ_b ”.

Thank you, corrected.

Figure 7a is supposed to be a plot against number of backbone entanglements, but on the figure, the top two curves are labelled as “100k-2x12-24k,” for regimes I/II, suggesting that all points on these curves are for a backbone of 100k. This does not seem possible, since the x axis is the length of backbone. Should the “100k” in the legend be replaced by “ $M_{w,b}$ ” to indicate that the backbone molecular weight is variable along the curve?

Thank you for your careful analysis. We adjusted the position of the text within the graph to make it clearer to which data point it is referring to.

In the summary, the “Rouse time” should be defined there, even if it is defined earlier (I’m not sure that it is).

We added a definition of the Rouse time in the summary. “When extended faster than the strain rate corresponding to the Rouse time all chains show strain hardening due to stretch due to monomer friction.”

In the response to my point (Reviewer 3) that “The scaling law of extensional viscosity with -0.5 exponent is not explained,” the authors give an explanation of their thinking. This explanation for the -0.5 exponent, rather than -1, should be mentioned in the text somewhere.

Thank you for the suggestion, we included the response to your point in the manuscript starting from line 372.

References

1. Affolter, S., Ritter, A. & Schmid, M. Interlaboratory Tests on Polymers by Differential Scanning Calorimetry (DSC): Determination of Glass Transition Temperature (T_g). *Macromol. Mater. Eng.* 286, 605–610 (2001).
2. Costanzo, S. et al. Strain Hardening of Unentangled Polystyrene Solutions in Fast Shear Flows. *Macromolecules* 55, 9206–9219 (2022).
3. Faust, L. et al. Comb and Branch-on-Branch Model Polystyrenes with Exceptionally High Strain Hardening Factor SHF $\text{greater than } 1000$ and Their Impact on Physical Foaming. *Macromol. Chem. Phys.* 224, 2200214 (2023).
4. Wagner, M. H. et al. The strain-hardening behaviour of linear and long-chain-branched polyolefin melts in extensional flows. *Rheol. Acta* 39, 97–109 (2000).
5. Wagner, M. H., Yamaguchi, M. & Takahashi, M. Quantitative assessment of strain hardening of low-density polyethylene melts by the molecular stress function model. *J. Rheol.* 47, 779–793 (2003).
6. Stadler, F. J., Kaschta, J., Münstedt, H., Becker, F. & Buback, M. Influence of molar mass distribution and long-chain branching on strain hardening of low density polyethylene. *Rheol. Acta* 48, 479–490 (2009).
7. Spitael, P. & Macosko, C. W. Strain hardening in polypropylenes and its role in extrusion foaming. *Polym Eng Sci* 44, 2090–2100 (2004).
8. Huang, G. et al. Strain Hardening Behavior of Poly(vinyl alcohol)/Borate Hydrogels. *Macromolecules* 50, 2124–2135 (2017).
9. Gabriel, C. & Münstedt, H. Strain hardening of various polyolefins in uniaxial elongational flow. *J. Rheol.* 47, 619–630 (2003).

Reviewers' Comments:

Reviewer #1:

Remarks to the Author:

The replies from the authors to my comments look fine with me. The descriptions in the revised manuscript are much clearer now.

There is one more point that I would like to ask the authors to clarify: On line 60 of the revised manuscript, the authors mentioned "the Rouse time" and ref.9 is cited there. Is the Rouse time here for a linear polymer? If so, does the length of the linear polymer correspond to the backbone of pom-pom (since the value of Rouse time depends on the molecular weight)?

Reviewer #3:

Remarks to the Author:

The paper has been further clarified. There are still some confusing aspects, especially regarding the "Rouse relaxation time" and the use of time scales limiting the validity of the theory. The paper is ready to publish after clarifying these remaining issues.

The following statement is still confusing. "Therefore, the pom-pom model is limited to strain rates smaller than the inverse of the arm relaxation time $1/\tau_a$ as well as the Rouse time τ_R because at Hencky strain rates $\dot{\epsilon} > 1/\tau_R$ all polymer chains show strain hardening due to monomer friction." So, the model is not valid in cases where there is strain hardening? I think what is meant is that the model is only valid when hardening is produced by the entangled arms, not by purely Rouse friction. If this is the case, it should be made clearer. Also, the idea that the pom-pom model is valid only for $\dot{\epsilon} > 1/\tau_b$ is not correct, at least for the original pom-pom model. The model at $\dot{\epsilon} < 1/\tau_b$ simply predicts Newtonian viscosity, so why would the model not be valid for that? Or are you only examining it for the case $\dot{\epsilon} > 1/\tau_b$? Care should be taken in saying in discussing the limits of the pom-pom model, to differentiate between limits on the authors' version of the model and the original pom-pom model or a more general version than considered by the authors.

Probably related to this is the confusing definition of the "Rouse time." The response to my inquiry says: "The Rouse time is the stretch relaxation time of linear polymers. In other words, at higher strain rates than the inverse of the Rouse time, even linear polymers show strain hardening. We included a reference for clarity. In the context of branched chains, the Rouse time corresponds to the whole molecule, including arms and backbone. We unified all references to Rouse time to increase clarity." Does the "Rouse time corresponds to the whole molecule" mean that it is the Rouse time of a linear molecule with the same molecular weight as the branched molecule? If so, this should be clarified. However, I note that later on, Eq. 13 gives a "Rouse time" for a branched polymer, where the formula clearly differs from that for a linear polymer. Thus, the statement that "The Rouse time is the stretch relaxation time of linear polymers" is contradicted by Eq. 13. Please clarify this.

In the answer to the question about Eq. 4, where the extensional viscosity on the left side is shown to be a function of strain rate and time, there still is no clear explanation why the right side shows no dependence on extension rate. Is the equation value for all extension rates above a critical value, in which case the time goes to t_q , which depends on the strain rate applied? And Eq. 5 is derived because a steady-state extensional viscosity is reached when $\epsilon_f \rightarrow \epsilon_q$? Is this correct? Again, more clarity on these points would help.

In response to the question about the time t_q , the authors say this: "Yes, affine stretching is assumed. The pom-pom model specifies $\tau_a < t < \tau_b$ for its validity." This is peculiar; affine stretching is dependent on having a high strain rate, not that the time is greater than a particular value or less than a particular value. If the extension rate is low, there will not be affine stretching of

the backbone at any times. Earlier it was said that the model is valid only when $1/\tau_a > \dot{\epsilon} > 1/\tau_b$. Isn't this the criterion you require for affine stretching? Why do you need a restriction on time?

The response to the question about τ_R , pom-pom clarifies what was done, but leaves the text itself confusing.

REVIEWER COMMENTS

Reviewer #1 (Remarks to the Author):

The replies from the authors to my comments look fine with me. The descriptions in the revised manuscript are much clearer now.

There is one more point that I would like to ask the authors to clarify: On line 60 of the revised manuscript, the authors mentioned “the Rouse time” and ref.9 is cited there. Is the Rouse time here for a linear polymer? If so, does the length of the linear polymer correspond to the backbone of pom-pom (since the value of Rouse time depends on the molecular weight)?

Thank you for your positive statement.

The Rouse time is related to the molecular weight (quadratic dependence for a linear polymer) and the polymer topology. Yes, the Rouse time mentioned in Ref. 9, now Ref. 10, is for linear polymers only, we added another one for branched polymers. In the case of a branched polymer like the pom-pom, the backbone and all branches contribute to the Rouse time of the whole molecule. To our knowledge, no relation for the Rouse time of a pom-pom and its molecular weight is known, that is why we used the equation for a comb as a first approximation.

Reviewer #3 (Remarks to the Author):

The paper has been further clarified. There are still some confusing aspects, especially regarding the “Rouse relaxation time” and the use of time scales limiting the validity of the theory. The paper is ready to publish after clarifying these remaining issues.

Thank you for your positive statement. In addition to the changes mentioned below, we included a list of all used symbols in the supporting information to increase the readability of the manuscript.

The following statement is still confusing. “Therefore, the pom-pom model is limited to strain rates smaller than the inverse of the arm relaxation time $1/\tau_a$ as well as the Rouse time τ_R because at Hencky strain rates $\dot{\epsilon} > 1/\tau_R$ all polymer chains show strain hardening due to monomer friction.” So, the model is not valid in cases where there is strain hardening? I think what is meant is that the model is only valid when hardening is produced by the entangled arms, not by purely Rouse friction. If this is the case, it should be made clearer.

Thank you. Yes, you are right. The strain hardening of pom-pom molecules, according to the pom-pom model, is produced by the friction of the branched points while the arms itself are relaxed ($\dot{\epsilon} < 1/\tau_a$). The model is indeed valid where strain hardening is produced by branch point friction and invalid at very high strain rates where strain hardening is produced significantly by monomer friction.

We clarified the statement to “The Rouse time of polymers τ_R indicates where strain hardening is only caused by monomer friction. This is the case if the applied elongational rate exceeds the inverse of the Rouse time. In the pom-pom model, strain hardening is the result of only branch point friction and at Hencky strain rates $\dot{\epsilon} > 1/\tau_R$ all polymer chains independent of their topology show strain hardening due to monomer friction.”

Also, the idea that the pom-pom model is valid only for $\dot{\epsilon} > 1/\tau_b$ is not correct, at least for the original pom-pom model. The model at $\dot{\epsilon} < 1/\tau_b$ simply predicts Newtonian viscosity, so why would the model not be valid for that? Or are you only examining it for the case $\dot{\epsilon} > 1/\tau_b$? Care should be taken in saying in discussing the limits of the pom-pom model, to differentiate between limits on the authors' version of the model and the original pom-pom model or a more general version than considered by the authors.

Thank you for your detailed review. You are right, the limit for slow strain rates results from the assumption of the Considère limit that the strain energy is stored as elastic energy and viscous components can be neglected. The pom-pom model is not limited at slow strain rates. The lower limit was removed in line 64 accordingly and the slow strain rate limit was moved to the introduction of the Considère limit in line 79. We added: “Rapid stretching is assumed for strain rates $\dot{\epsilon} > 1/\tau_s$, where the viscous components can be neglected, and the strain energy is stored as elastic energy.”

Probably related to this is the confusing definition of the “Rouse time.” The response to my inquiry says: “The Rouse time is the stretch relaxation time of linear polymers. In other words, at higher strain rates than the inverse of the Rouse time, even linear polymers show strain hardening. We included a reference for clarity. In the context of branched chains, the Rouse time corresponds to the whole molecule, including arms and backbone. We unified all references to Rouse time to increase clarity.” Does the “Rouse time corresponds to the whole molecule” mean that it is the Rouse time of a linear molecule with the same molecular weight as the branched molecule? If so, this should be clarified. However, I note that later on, Eq. 13 gives a “Rouse time” for a branched polymer, where the formula clearly differs from that for a linear polymer. Thus, the statement that “The Rouse time is the stretch relaxation time of linear polymers” is contradicted by Eq. 13. Please clarify this.

Thank you for pointing this out, we understand that the original phrasing of the sentence was misleading, as we meant that for linear polymers, the Rouse time equals the stretch relaxation time. We did not mean to say that the Rouse time is a quantity only for linear polymers. The Rouse time has different dependencies on molecular weight and topology for linear and branched polymers.

The “Rouse time corresponds to the whole molecule” means that all parts (all arms and the backbone) of a branched polymer contribute to its Rouse time.

As discussed in response to earlier questions and in the manuscript, the Rouse time is dependent on the molecular weight and the topology of the polymer. For linear chains, the relation $\tau_R = s^2 * \tau_e$ is established, which depends only on its number of entanglements s in the linear chain, i.e. its molecular weight. The Rouse time of branched polymers is unknown, for combs Lentzakis and coworkers developed Eq. 13, $\tau_{R,br.} = \tau_e s_b (s_b + 2q s_a)$. For the pom-pom topology, no relation of the Rouse time on $M_{w,a}$, $M_{w,b}$ and q is known to us.

Thank you again for catching this ambiguous phrasing.

In the answer to the question about Eq. 4, where the extensional viscosity on the left side is shown to be a function of strain rate and time, there still is no clear explanation why the right side shows no dependence on extension rate. Is the equation value for all extension rates above a critical value, in which case the time goes to t_q , which depends on the strain rate applied?

Yes, indeed Eq. 4 is only valid for rapid stretching as described from line 77f onwards. Therefore, all strain rates under rapid stretching reach $\varepsilon_q = t_q \dot{\varepsilon}$. The right side of Eq. 4 depends also on the strain rate.

$$\log(\eta_E^+(t, \dot{\varepsilon})/\eta_0)|_{\varepsilon_f \rightarrow \varepsilon_q} = \log\left(\frac{3q^2}{\varepsilon_q}\right) + \log\left(\frac{t_q}{\tau_b}\right)$$

To reach ε_q at t_q , a certain strain rate is needed, so $\varepsilon_q = t_q \dot{\varepsilon}$, i.e. it depends on the strain rate. Additionally, t_q can also be expressed in terms of $\dot{\varepsilon}$ and ε_q , yielding

$$\log(\eta_E^+(t, \dot{\varepsilon})/\eta_0)|_{\varepsilon_f \rightarrow \varepsilon_q} = \log\left(\frac{3q^2}{t_q \dot{\varepsilon}}\right) + \log\left(\frac{\varepsilon_q}{\dot{\varepsilon} \tau_b}\right)$$

Where also the right side depends clearly on the strain rate. When ε_q is reached, the maximum stress on the backbone is obtained and the arms will be withdrawn into the backbone tube resulting in a constant stress.

And Eq. 5 is derived because a steady-state extensional viscosity is reached when $\varepsilon_f \rightarrow \varepsilon_q$? Is this correct? Again, more clarity on these points would help.

Due to the application of $\varepsilon_f \rightarrow \varepsilon_q$ onto the term $\eta_E^+(t, \dot{\varepsilon})/\eta_0$ the time dependent tensile stress growth coefficient transforms into the time-independent steady state extensional viscosity, which is independent of the measurement time, as Eq. 5 can be transformed into $\frac{\eta_E}{\eta_0} = \frac{3q^2}{\tau_b} \cdot \frac{t_q}{\varepsilon_q} = \frac{3q^2}{\tau_b} \cdot \frac{1}{\dot{\varepsilon}}$ and becomes only a function of the applied strain rate. This is also logic, as the steady state value of the tensile stress growth coefficient must be a function of the applied strain rate but is by definition independent of the measurement time. The second part of Eq. 5 was added to the manuscript.

In response to the question about the time t_q , the authors say this: “Yes, affine stretching is assumed. The pom-pom model specifies $\tau_a < t < \tau_b$ for its validity.” This is peculiar; affine stretching is dependent on having a high strain rate, not that the time is greater than a particular value or less than a particular value. If the extension rate is low, there will not be affine stretching of the backbone at any times. Earlier it was said that the model is valid only when $1/\tau_a > \dot{\varepsilon} > 1/\tau_b$. Isn't this the criterion you require for affine stretching? Why do you need a restriction on time?

Yes, you are right, the decisive quantity is the strain rate. The limits of the Considère limit applied to the pom-pom model are better expressed with strain rate and stretching of the backbone is observed at $1/\tau_a < \dot{\varepsilon} < 1/\tau_b$.

The response to the question about τ_R , pom-pom clarifies what was done, but leaves the text itself confusing.

We added to the manuscript in line 335f: “To this point we are unaware of any exact equation for the prediction of the Rouse time of a pom-pom. We estimate that the Rouse time of a comb should be slightly larger than the Rouse time of a pom-pom due to its increased inertia at the chain ends.”

Reviewers' Comments:

Reviewer #3:

Remarks to the Author:

The paper is ready to publish. I have only a couple of optional comments.

The statement "The Rouse time of polymers indicates where strain hardening is only caused by monomer friction" would be better rendered: "The Rouse time of a polymer, linear or branched, is the longest relaxation time of the molecule in the absence of entanglement effects; i.e., governed only by monomeric friction." The inverse of the Rouse time of a polymer is thus the strain rate above which strain hardening can occur by monomer friction alone."

I understand the response to my question about Eq. 4, but the reader will understand better if, after Eq. 4, a statement is made that "Here $t_q = \epsilon_q / \dot{\epsilon}$, and thus is dependent on strain rate."

Answers to the Reviewers

Reviewer #3 (Remarks to the Author):

The paper is ready to publish. I have only a couple of optional comments.

Thank you for your positive statement.

The statement “The Rouse time of polymers indicates where strain hardening is only caused by monomer friction” would be better rendered: “The Rouse time of a polymer, linear or branched, is the longest relaxation time of the molecule in the absence of entanglement effects; i.e., governed only by monomeric friction.” The inverse of the Rouse time of a polymer is thus the strain rate above which strain hardening can occur by monomer friction alone.”

Thank you, we adapted the suggested statement in our manuscript.

I understand the response to my question about Eq. 4, but the reader will understand better if, after Eq. 4, a statement is made that “Here $t_q = \epsilon_q / \dot{\epsilon}$, and thus is dependent on strain rate.”

Thank you, we included the suggested relation.